# CoT-lized Diffusion: Let's Reinforce T2I Generation Step-by-step

**Zheyuan Liu**[1,3*]    **Munan Ning**[1*]    **Qihui Zhang**[1,3]    **Shuo Yang**[1]    **Zhongrui Wang**[1]
**Yiwei Yang**[4]    **Xianzhe Xu**[2,3]    **Yibing Song**[2,3]    **Weihua Chen**[2,3†]    **Fan Wang**[3]    **Li Yuan**[1†]

[1]Shenzhen Graduate School, Peking University    [2]Hupan Lab
[3]DAMO Academy, Alibaba Group    [4]Shanghai Jiao Tong University

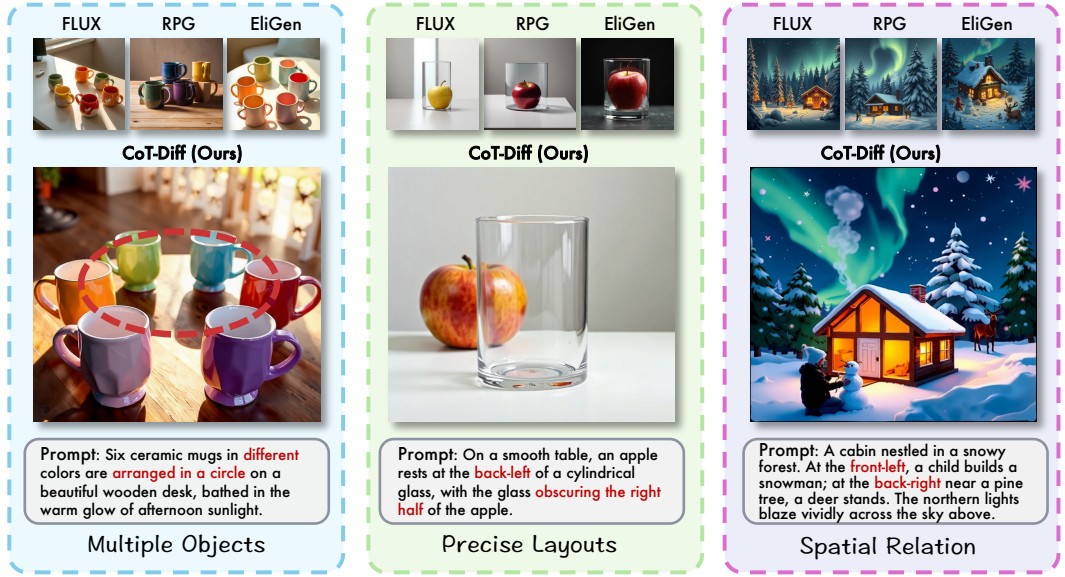

Figure 1: Comparison of generated images across three challenging spatial scenarios: multiple-object configuration (left), precise layout (middle), and complex spatial relations (right). CoT-Diff (ours) achieves significantly better spatial alignment than FLUX [5], RPG [47], and EliGen [48], faithfully following 3D-aware instructions in the prompt (highlighted in red).

## Abstract

Current text-to-image (T2I) generation models struggle to align spatial composition with the input text, especially in complex scenes. Even layout-based approaches yield suboptimal spatial control, as their generation process is decoupled from layout planning, making it difficult to refine the layout during synthesis. We present **CoT-Diff**, a framework that brings step-by-step CoT-style reasoning into T2I generation by tightly integrating Multimodal Large Language Model (MLLM)-driven 3D layout planning with the diffusion process. CoT-Diff enables layout-aware reasoning inline within a single diffusion round: at each denoising step, the MLLM evaluates intermediate predictions, dynamically updates the 3D scene layout, and continuously guides the generation process. The updated layout is

---

\* Equal contribution    † Corresponding author
{liuzy2233, munanning}@gmail.com
kugang.cwh@alibaba-inc.com yuanli-ece@pku.edu.cn

39th Conference on Neural Information Processing Systems (NeurIPS 2025).

converted into semantic conditions and depth maps, which are fused into the diffusion model via a condition-aware attention mechanism, enabling precise spatial control and semantic injection. Experiments on 3D Scene benchmarks show that CoT-Diff significantly improves spatial alignment and compositional fidelity, and outperforms the state-of-the-art method by 34.7% in complex scene spatial accuracy, validating the effectiveness of this entangled generation paradigm.

# 1 Introduction

Diffusion models have shown remarkable capabilities in generating high-quality and diverse images from textual descriptions [15, 11, 22]. However, these models lack explicit structural reasoning abilities [51, 45, 52]. When tasked with complex scenes involving multiple objects, precise layouts, and spatial relations, current models still face significant limitations in controllability, thereby limiting their applicability to structured generation tasks.

Recent works have introduced layout-aware conditions into the diffusion process, incorporating points, scribbles, 2D bounding boxes or semantic masks to enhance entity-level spatial control [49, 27, 21, 50, 40]. Yet, most of these methods [23, 43] adopt a decoupled paradigm where layout planning is conducted before generation, without any feedback from the image synthesis process. Such static pipelines lack the ability to refine spatial arrangements, and thus often struggle to model fine-grained structure in complex scenes. Some recent 3D-aware approaches [4, 10, 39] support explicit object placement in space, but typically rely on manually crafted layouts and exhibit limited generalization in automated generation settings. These limitations highlight the need for a unified framework that can reason about 3D structure from text and dynamically control layout during generation—a goal we aim to achieve in this work.

We present **CoT-Diff**, a 3D-aware text-to-image framework for stepwise, fine-grained spatial control over complex visual scenes as shown in Figure 1. Unlike prior approaches that perform layout planning as a static pre-process, CoT-Diff tightly entangles a multimodal large language model (MLLM) with a diffusion model, enabling inline scene reasoning and generation within a single sampling trajectory. Given a text prompt, CoT-Diff first uses the MLLM to plan an initial 3D scene layout including entity descriptions, spatial positions, and size estimates. Then, at each denoising step, the intermediate predicted image is passed back to the MLLM, which re-evaluates alignment with the input text and adjusts the layout plan accordingly, modifying selected entities' attributes.

The updated layout is immediately transformed into two types of spatial conditions: a **semantic layout**, encoded from the global prompt and the entity local prompts, and a **depth map**, rendered from the relative depth of each object given the scene geometry and camera view. These spatial priors are encoded and injected into the latent features of the pretrained diffusion model to guide the generation process with precise semantic and geometric constraints. In contrast to iterative editing methods that regenerate images after layout revision, CoT-Diff performs all reasoning and control within a single diffusion round, avoiding redundant denoising while maintaining stepwise structural consistency. To support multi-source conditioning, we introduce separate semantic and depth control branches, implemented via lightweight Low-Rank Adaptation (LoRA) modules [16], and propose a **condition-aware attention** mechanism that spatially disentangles different modalities and injects control signals only into relevant image regions.

To support structured layout supervision for training, we construct an automatically annotated 3D layout dataset based on EliGen [48] and LooseControl [4]. Building upon the original annotations of global prompts, per-entity descriptions, and 2D bounding boxes, we incorporate monocular depth estimation and segmentation models to recover entity-level depth and generate 3D bounding boxes via geometric fitting. To validate CoT-Diff, we present a new benchmark, dubbed **3DSceneBench**, consisting of diverse and complex compositions of spatial relationship. On 3DSceneBench and two existing T2I benchmarks [34, 17], CoT-Diff outperforms state-of-the-art diffusion baselines, including FLUX and RPG [47], by up to 34.7% in terms of complex scene spatial accuracy.

This paper makes the following contributions:

- We propose **CoT-Diff**, a novel framework that tightly couples multimodal large language models (MLLMs) with diffusion models for stepwise 3D-aware image generation.

- We introduce an inline layout reasoning mechanism, where MLLM dynamically updates the 3D layout at each denoising step, enabling CoT-style spatial control within a single diffusion round.
- We design disentangled spatial control modules that convert each layout plan into semantic masks and depth maps, which are selectively injected via a condition-aware attention mechanism.
- We build an automatically annotated 3D-aware layout dataset, enabling entity-level layout supervision and structured evaluation of spatial consistency.

## 2 Related Work

### 2.1 Text-to-Image Diffusion Models

Recent advances in diffusion models have greatly improved the quality and diversity of text-to-image (T2I) generation [15, 37]. Latent Diffusion Models (LDMs) [35, 31] further enhance efficiency by operating in a compressed latent space. Modern systems like SD3 [11], Hunyuan-dit [22], and FLUX [5] adopt transformer-based backbones such as DiT [30] and leverage pretrained text encoders (e.g., CLIP [32], T5 [33]) to map text into rich visual content. However, these models treat text as a static global condition and lack explicit control over spatial composition. When dealing with complex scenes containing multiple entities and spatial relationships [49, 27], they often fail to reason about relative positioning, depth, or occlusion. This limitation has led to efforts incorporating layout or structure-based conditions into the generation pipeline.

### 2.2 Layout-guided Image Generation

To improve spatial controllability, layout-guided methods introduce object-level structure into T2I generation. Training-based approaches [21, 40, 50] inject layout or mask annotations into attention modules during training. EliGen [48] pushes this further by using fine-grained entity-level prompts. Other methods avoid retraining by designing plug-and-play mechanisms. For example, MultiDiffusion [3] applies region-specific denoising followed by fusion; RPG [47] and RAG-Diffusion [9] use resize-and-concatenate schemes to combine regional latents. While these methods improve control, they mainly operate in 2D and lack true 3D spatial reasoning. More recent works explore 3D-aware generation [4, 10], but often rely on manually defined layouts and still lag behind top-performing 2D models in visual quality. These limitations motivate automated 3D planning and controllable generation directly from text.

### 2.3 MLLM-Grounded Diffusion

Large Language Models (LLMs) exhibit strong reasoning capabilities through techniques such as chain-of-thought prompting [42] and reinforcement fine-tuning [13]. Their multimodal counterparts (MLLMs) [25, 53, 36] extend this ability to visual-linguistic understanding, enabling structured scene interpretation from natural language. LayoutGPT [12] and LayoutVLM [38] use MLLMs to convert captions into 2D or 3D layouts, which can support downstream image generation. RPG [47] incorporates layout planning into diffusion synthesis using regional generation. Despite these advances, existing MLLM-guided diffusion systems [41, 24] focus primarily on static layout generation and 2D spatial arrangements. They lack iterative reasoning or full 3D understanding, which limits their performance in dense or physically plausible scenes. In contrast, Plan2Pix uses MLLMs for dynamic 3D planning and integrates a feedback loop that adjusts layouts based on generation outcomes, enabling more accurate and semantically aligned control over complex scenes.

## 3 Method

### 3.1 Overview of CoT-Diff Framework

As illustrated in Figure 2, CoT-Diff enables step-by-step coupling between a MLLM and a diffusion model, allowing layout reasoning and image generation to proceed in tandem. At each step, the updated 3D layout is converted into semantic masks and depth maps, which encode spatial priors and are injected into the diffusion model via LoRA modules and a condition-aware attention mechanism for disentangled and precise control.

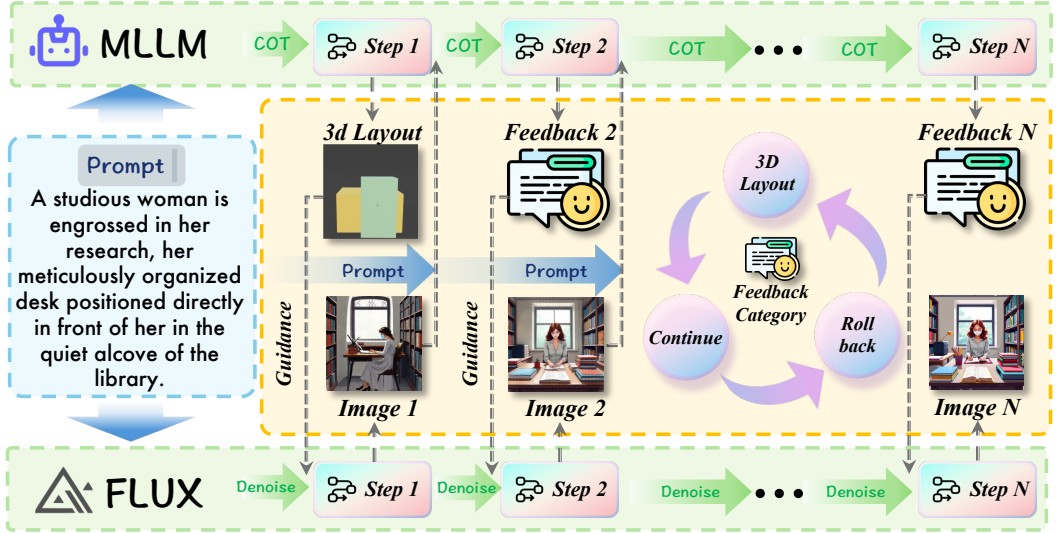

Figure 2: Overview of the **CoT-Diff** framework. Given a prompt, an MLLM first plans a 3D scene layout and then collaborates with the diffusion model in a step-by-step manner. At each denoising step, the MLLM inspects intermediate predictions, refines the layout through CoT-style reasoning, and provides updated guidance to the diffusion model.

## 3.2 MLLM-guided 3D Scene Planning with Stepwise optimization

We propose a CoT-lized reasoning mechanism, where a MLLM collaborates with the diffusion model to perform stepwise scene planning and optimization. Unlike previous approaches that decouple layout from generation, our framework tightly integrates MLLM-driven layout reasoning into the sampling trajectory, enabling dynamic updates to the 3D plan during generation.

**Initial 3D Scene Planning.** We use a MLLM to parse the input prompt $p$ and generate a structured 3D scene plan. The model identifies key entities, estimates their physical attributes, and determines their spatial arrangement in the scene. The MLLM first identifies $k$ salient entities from the prompt, forming an entity set:

$$E = \{e_j\}_{j=1}^k. \tag{1}$$

For each entity $e_j \in E$, the MLLM generates a local text prompt $p_j$ describing its appearance and attributes. It also predicts a plausible 3D size $\text{size}_j = (l_j, w_j, h_j)$ and a spatial coordinate $\text{pos}_j = (x_j, y_j, z_j)$. The overall output is represented as:

$$f_{\text{MLLM}}(e_j|p) \rightarrow (p_j, \text{size}_j, \text{pos}_j). \tag{2}$$

This planning is based on entity semantics and spatial relationships implied in the prompt, guided by the commonsense knowledge embedded in the MLLM. The final 3D scene plan is:

$$\mathcal{SP} = (p, \{(p_j, \text{size}_j, \text{pos}_j)\}_{j=1}^k). \tag{3}$$

**Stepwise Feedback-based Refinement.** While the initial 3D layout provides strong spatial guidance, discrepancies may still arise during generation. To address this, CoT-Diff introduces an iterative optimization loop driven by MLLM feedback. At each timestep $t$, the model predicts a clean image from noisy latent $z_t$ via:

$$\hat{x}_{0|t} = z_t - t \cdot v_t. \tag{4}$$

The MLLM evaluates $\hat{x}_{0|t}$ by comparing it against both the input prompt $p$ and the current plan $\mathcal{SP}$. If misalignment is detected, the MLLM proposes refined attributes for selected entities (e.g., $\text{size}_j$ or $\text{pos}_j$), producing an updated plan:

$$\mathcal{SP}' = \text{Refine}(\mathcal{SP}, \hat{x}_{0|t}, p), \tag{5}$$

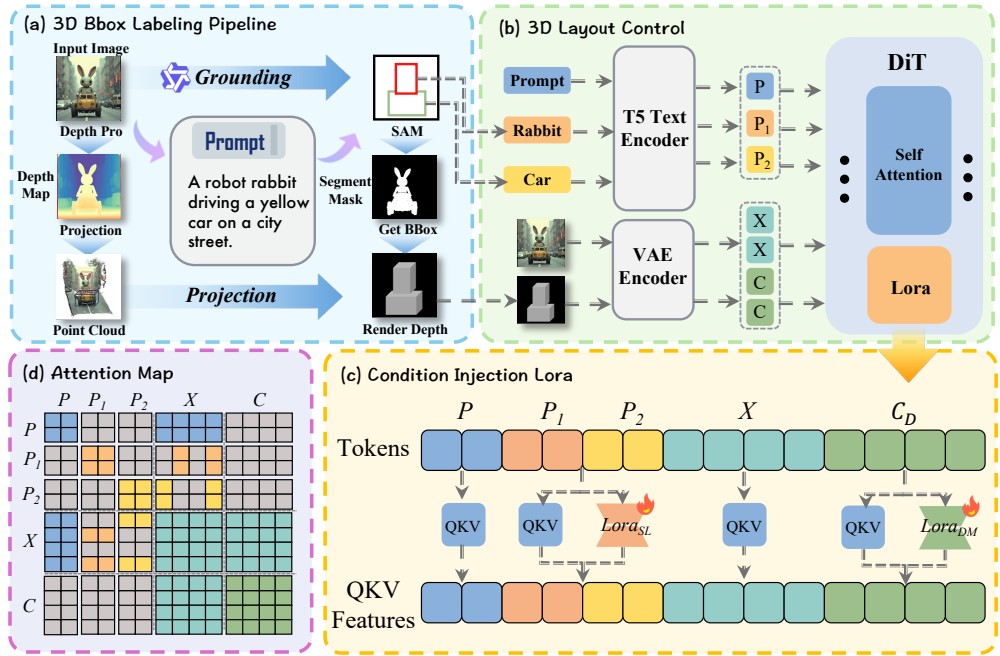

Figure 3: Illustration of 3D layout conditioning and condition-aware attention. (a) 3D bounding boxes are automatically labeled from input images using depth estimation, segmentation, and projection. (b) Text and spatial inputs are encoded into semantic and geometric conditions via T5 and VAE encoders. (c) Condition Injection LoRA activates modality-specific branches during QKV projection. (d) A learned attention mask enforces condition-wise separation and spatially localized injection.

which is used to reguide the denoise process at timestep $t$. This predict-evaluate-refine cycle progressively enhances spatial and semantic consistency. Evaluation continues for a maximum of 5 steps or until the MLLM deems the layout correct.

Unlike prior methods that rely on multi-round generation, our framework performs planning and optimization within a single diffusion process, enabling efficient and interpretable 3D-aware generation.

### 3.3 3D Layout Control

Given the 3D scene plan $\mathcal{SP}$ from Section 3.2, we extract two types of conditional signals: (i) semantic layout and (ii) depth maps. These are jointly injected into the diffusion model via a condition-aware attention mechanism, enabling precise control over the spatial and semantic structure.

**Semantic Layout Condition.** This module focuses on injecting global and entity-specific semantics into corresponding regions of the image. As illustrated on top-left of Figure 3(b), we extract global embedding $P$ and local embeddings $\{P_j\}$ from the global prompt $p$ and local prompts $\{p_j\}$ using a T5 encoder, and train a Semantic LoRA module, denoted $LoRA_{SL}$, to control this injection. The model is optimized using the following conditional flow matching loss:

$$L_{SL} = \mathbb{E}_{t,z,\epsilon} \left[ \| v_\Theta(z, t, P, P_1, \dots, P_k) - u_t(z|\epsilon) \|^2 \right], \quad (6)$$

where $v_\Theta$ is the velocity field predicted by the model augmented by $LoRA_{SL}$.

**Depth Map Condition.** This module focuses on extracting depth information to guide spatial structure and object positioning, as illustrated on the bottom-left of Figure 3(b). Starting from the 3D scene plan $\mathcal{SP}$, we obtain the set of 3D bounding boxes $\{B_j\}$ and render a corresponding depth map $D$. This depth map is then encoded using the DiT VAE to produce the latent depth condition $C_D$. We inject $C_D$ into the diffusion model via a dedicated Depth LoRA module, $LoRA_{DM}$, which is optimized using the following conditional flow matching loss:

$$L_{DM} = \mathbb{E}_{t,z,\epsilon} \left[ \| v_\Phi(z, t, C_D, P_{emb}) - u_t(z|\epsilon) \|^2 \right], \quad (7)$$

where $v_\Phi$ is the velocity field predicted by the model augmented by $LoRA_{SL}$ and $LoRA_{DM}$, $u_t(z|\epsilon)$ is the target velocity field, and $P_{emb}$ is the all text prompt embedding.

**Condition-Aware Attention Mechanism.** To integrate semantic and depth conditions without cross-interference, we design a Condition Attention mechanism that selectively controls how different condition types attend to the image tokens, ensuring spatial alignment while avoiding modality entanglement. We begin by constructing a unified token sequence by concatenating the semantic tokens, image tokens, and depth tokens, as illustrated in Figure 3(c):

$$S = [P, P_1, \ldots, P_k, C_D, X], \tag{8}$$

where $P$ is the global prompt token, $\{P_j\}_{j=1}^k$ are local prompt tokens for each entity, $C_D$ is the depth condition token, and $X$ denotes the image latent tokens.

To guide attention computation, we construct a binary attention mask $M \in \{0,1\}^{|S| \times |S|}$ that controls which tokens are allowed to attend to one another, as illustrated in Figure 3(d). The mask is designed to satisfy three principles: (i) tokens from different condition sources are mutually isolated to prevent cross-condition interference, i.e.,

$$M(a,b) = 0 \quad \text{if } a,b \in \{P_1, \ldots, P_k, C_d\},\, a \neq b; \tag{9}$$

(ii) all tokens are allowed to self-attend, i.e., $M(a,a) = 1$ for all $a \in S$; and (iii) interactions between condition tokens and image latents are modulated by spatial masks. Specifically, global conditions $P_0$ and $C_d$ attend to all latents:

$$M(P_0, X) = M(X, P_0) = M(C_d, X) = M(X, C_d) = 1, \tag{10}$$

while each local prompt $P_j$ is restricted to its associated region:

$$M(P_j, X) = M(X, P_j) = \text{patchify}(m_j), \tag{11}$$

where $m_j$ is a 2D mask derived by projecting the 3D bounding box $B_j$ onto the image plane. With this mask $M$, the attention output is computed as:

$$\text{Attn}(Q,K,V) = \text{Softmax}\left(\frac{QK^\top}{\sqrt{d_k}} + \log M\right)V. \tag{12}$$

This formulation allows the model to attend over spatially relevant regions for each condition while avoiding semantic-depth entanglement and cross-entity confusion.

### 3.4 3D-Aware Dataset Construction Pipeline

As shown in Figure 3(a), we construct a 3D-aware dataset which automatically generates 3D bounding boxes for each object. This dataset includes global prompts, local prompts, and 3D bounding boxes.

We generate depth maps using a monocular depth estimator (Depth Pro) [6], and apply the Segment Anything Model (SAM) [20] with 2D bounding boxes to extract segmentation masks. For each object, we extract its depth region and back-project it into a 3D point cloud: $\mathbf{C} = \{(x, y, d_i(x,y)) \mid (x,y) \in m_i\}$, where $d_i$ denotes the depth values. We then fit an oriented 3D bounding box with minimal volume: $\min_{\mathbf{B}} \text{Volume}(\mathbf{B}) = \text{fit}(\mathbf{C})$.

Finally, we re-project the 3D bounding boxes onto the image space and render depth maps, ensuring accurate and consistent 3D representations.

## 4 Experiments

### 4.1 Experimental Setup

**Implementation Details.** Our method is implemented using Gemini 2.5 Pro as the default multimodal language model and FLUX.1-schnell [5] as the diffusion model. We perform all inferences with 20 denoising steps and use 5 different random seeds for fairness. The LoRA scale is set to 1. All models are implemented in PyTorch and executed on NVIDIA A100 GPUs.

**Baselines.** We compare our method with two categories of existing approaches: (1) *Pre-trained T2I diffusion models*, including SD1.5 [35], SDXL [31], PixArt [9], and FLUX-schnell [5]; (2) *Layout-controlled methods*, including RPG [47], EliGen [48], and Loose Control (LC) [4], which incorporate external layout representations to guide spatial placement of objects. All baselines are evaluated under consistent settings using their official or publicly available implementations.

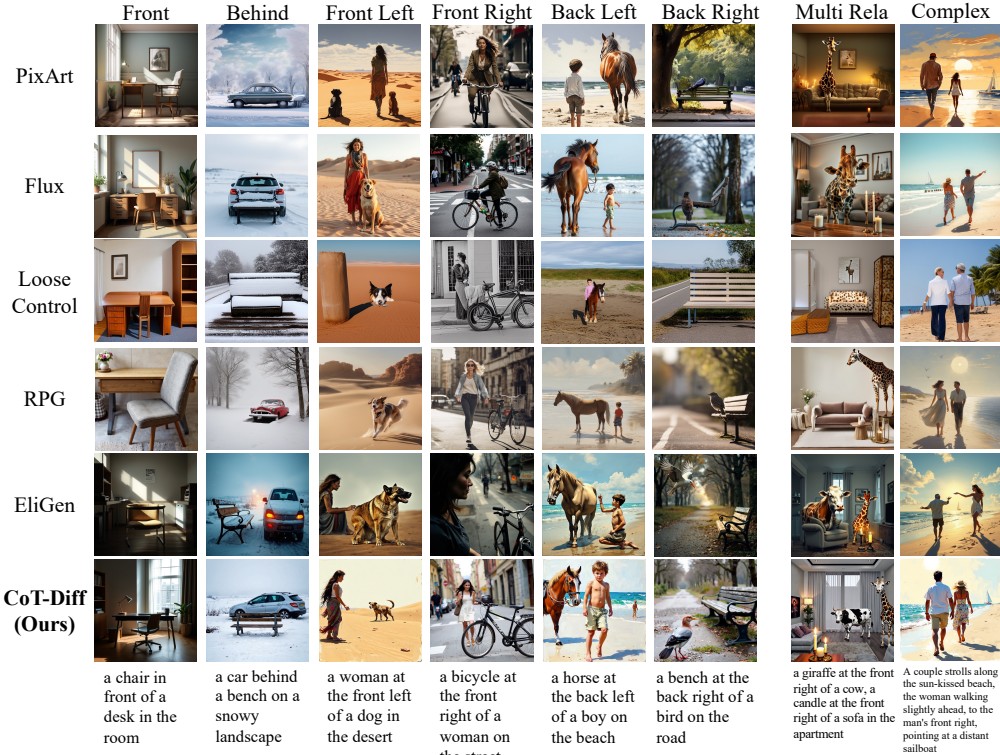

|  | Front | Behind | Front Left | Front Right | Back Left | Back Right | Multi Rela | Complex |
|---|---|---|---|---|---|---|---|---|
| PixArt | | | | | | | | |
| Flux | | | | | | | | |
| Loose Control | | | | | | | | |
| RPG | | | | | | | | |
| EliGen | | | | | | | | |
| **CoT-Diff (Ours)** | | | | | | | | |
| | a chair in front of a desk in the room | a car behind a bench on a snowy landscape | a woman at the front left of a dog in the desert | a bicycle at the front right of a woman on the street | a horse at the back left of a boy on the beach | a bench at the back right of a bird on the road | a giraffe at the front right of a cow, a candle at the front right of a sofa in the apartment | A couple strolls along the sun-kissed beach, the woman walking slightly ahead, to the man's front right, pointing at a distant sailboat |

Figure 4: Qualitative comparison of CoT-Diff and baselines across spatial relation categories.

**Datasets.** We evaluate our CoT-Diff with one new dataset, **3DSceneBench**, which we design to validate the generation ability in complex 3D scenes, and two existing datasets, DVMP [34] and T2I-CompBench [17, 18], for general compositional image generation. As shown in Table 1, 3DSceneBench includes more diverse and spatially grounded relationships compared to existing benchmarks. Further evaluation details can be found in the Appendix G.

Table 1: Dataset statistics. %Complex is the rate of spatial complex prompts evaluated by GPT4. (See Appendix F).

| Datasets | 3DSceneBench | DVMP | T2I-CompBench |
|---|---|---|---|
| # Prompts | 800 | 200 | 2400 |
| % Complex | 95.3 | 23.0 | 27.6 |

1. **3DSceneBench:** We design the 3DSceneBench dataset to cover a wide range of realistic 3D spatial relationships. All prompts are categorized into two major types based on relational complexity: **Basic Relation** and **Hard Relation**. The Basic Relation group includes six canonical spatial configurations: Front, Behind, Front Left, Front Right, Back Left, and Back Right (e.g., "a {object1} in front of a {object2} in the {scene}"). The Hard Relation group contains more challenging compositions, including *Multi-Relation* (e.g., "a {object1} {relation1} a {object2}, a {object3} {relation2} a {object4} in the scene") and *Complex* prompts to reflect complex 3D spatial logic. We generate 100 prompts for each of the eight relation types, resulting in a total of 800 prompts. All prompts are manually validated to ensure semantic clarity and spatial correctness. Constructing process of 3DSceneBench is detailed in Appendix E.

2. **DVMP:** This dataset is built by randomly pairing 38 objects with 26 attributes (including 13 colors), covering both single-object and multi-object prompts, with 100 examples for each setting.

3. **T2I-CompBench:** This benchmark contains multi-object prompts with relatively common compositions, serving as a baseline for evaluating spatial and semantic consistency.

### 4.2 Main Results of CoT-Diff

**3DSceneBench.** We first present a qualitative comparison as shown in Figure 4. Traditional diffusion models (SD1.5, SDXL, PixArt, FLUX) perform poorly due to the absence of explicit layout

Table 2: Text-to-image alignment performances of CoT-Diff and other baselines on the **3DSceneBench** dataset. The best values are in blue and the second best values are in green .

| Models | Basic Rela | | | | | | Hard Rela | |
|---|---|---|---|---|---|---|---|---|
| | Front | Behind | Front Left | Front Right | Back Left | Back Right | Multi Rela | Complex |
| SD1.5 | 32.8 | 36.8 | 26.5 | 28.6 | 34.1 | 33.7 | 21.8 | 34.0 |
| SDXL | 39.3 | 43.9 | 36.7 | 35.6 | 35.4 | 40.2 | 27.7 | 34.9 |
| PixArt | 39.4 | 43.7 | 33.8 | 32.2 | 35.4 | 36.4 | 27.0 | 37.65 |
| Flux | 48.0 | 45.9 | 40.7 | 40.8 | 33.3 | 37.4 | 35.2 | 40.5 |
| RPG | 40.5 | 38.2 | 37.1 | 39.8 | 38.8 | 42.1 | 25.2 | 36.5 |
| EliGen | 43.6 | 42.9 | 48.4 | 52.5 | 46.7 | 39.9 | 34.8 | 40.6 |
| LC | 25.3 | 28.5 | 26.9 | 29.0 | 35.5 | 27.0 | 12.7 | 23.7 |
| **CoT-Diff** | 54.9 | 55.2 | 66.4 | 69.0 | 64.2 | 64.5 | 48.7 | 50.8 |

Table 3: T2I alignment performance of CoT-Diff and baselines on DVMP and T2I-CompBench.

| Models | DVMP | | T2I-CompBench | | | | | |
|---|---|---|---|---|---|---|---|---|
| | Single | Multi | Color | Shape | Texture | Spatial | Non-Spatial | Complex |
| SD1.5 | 65.0 | 37.8 | 37.5 | 38.8 | 44.1 | 09.5 | 31.2 | 30.8 |
| SDXL | 76.8 | 59.0 | 58.8 | 46.9 | 53.0 | 21.3 | 31.2 | 32.4 |
| PixArt | 73.5 | 44.0 | 66.9 | 49.3 | 64.8 | 20.6 | 32.0 | 34.3 |
| Flux | 66.8 | 72.5 | 74.1 | 57.2 | 69.2 | 28.6 | 31.3 | 37.0 |
| RPG | 74.0 | 30.0 | 64.1 | 45.8 | 56.6 | 48.8 | 30.2 | 44.3 |
| EliGen | 78.7 | 78.5 | 72.8 | 58.7 | 66.8 | 53.6 | 31.2 | 43.9 |
| LC | 51.6 | 45.7 | 27.9 | 29.1 | 26.4 | 26.7 | 30.2 | 28.1 |
| **CoT-Diff** | 80.9 | 78.8 | 78.1 | 61.1 | 69.1 | 55.8 | 31.3 | 50.0 |

guidance. While layout-guided methods like RPG and EliGen incorporate positional control, they still demonstrate limited capability in modeling complex depth relationships. In contrast, CoT-Diff accurately synthesizes the specified spatial relationships by tightly coupling layout reasoning and generation, dynamically refining the scene plan, and injecting spatial guidance at each denoising step.

For quantitive experiment, Table 2 shows the spatial scene alignment accuracy of CoT-Diff compared to all the baselines on the 3DSceneBench dataset. Overall, CoT-Diff consistently outperforms all baselines across all spatial relationship categories. Numerically, CoT-Diff achieves the highest alignment scores in both basic and complex relational categories, with improvements ranging from +10.2% to +22.4% over the best baselines.

**DVMP and T2I-CompBench.** Table 3 summarizes the T2I alignment performance of CoT-Diff and baseline models on DVMP and T2I-CompBench. On DVMP, CoT-Diff achieves the highest scores across all attribute-binding categories—Color, Shape, and Texture—outperforming the strongest baseline (EliGen) by 2.3% to 5.3%. This confirms that semantic-aware generation in CoT-Diff strengthens consistency between visual attributes and textual descriptions. In contrast, traditional diffusion models such as SD1.5, SDXL, and PixArt show clearly inferior performance due to the lack of layout or semantic control. On T2I-CompBench, CoT-Diff also leads in most categories, with notable improvements in Spatial (+2.2%) and Complex (+6.7%) over EliGen. While EliGen incorporates 2D layout planning, its performance degrades in cases involving deeper relational reasoning. In contrast, CoT-Diff demonstrates more robust behavior by leveraging structure-aware generation guided by semantic planning. Overall, CoT-Diff excels not only in 3D-structured scenes but also generalizes well to traditional T2I alignment tasks.

### 4.3 Ablation Study of CoT-Diff

**Component Analysis.** We conduct an ablation study to evaluate the contribution of each core component in CoT-Diff, starting from the FLUX base model. As shown in Table 4, adding semantic layout guidance significantly improves multi relationship (+4.9) and complex scene (+3.2) accuracy by providing better object placement control. Introducing depth control further enhances performance, especially in multi relationship (+1.6) and complex (+4.8) settings, by resolving occlusion and ensuring accurate object ordering. Finally, the full CoT-Diff model, with optimized settings, achieves

Table 4: Ablation study of CoT-Diff. We progressively add layout, depth, and optimization components on top of the FLUX base.

| Model Variant | Front | Behind | Front Left | Front Right | Back Left | Back Right | Multi Rela | Complex |
|---|---|---|---|---|---|---|---|---|
| FLUX | 48.0 | 45.9 | 40.7 | 40.8 | 33.3 | 37.4 | 35.2 | 40.5 |
| + Semantic Layout | 44.5 | 48.0 | 56.9 | 58.0 | 51.9 | 50.6 | 40.1 | 43.7 |
| + Depth Map | 51.3 | 51.2 | 62.2 | 61.4 | 59.9 | 56.7 | 41.7 | 48.5 |
| + Optim (Full CoT-Diff) | **54.9** | **55.2** | **66.4** | **69.0** | **64.2** | **64.5** | **48.7** | **50.8** |

Table 5: CoT-Diff with different MLLMs.

| Models | Front | Behind | Front Left | Front Right | Back Left | Back Right |
|---|---|---|---|---|---|---|
| Flux | 48.0 | 45.9 | 40.7 | 40.8 | 33.3 | 37.4 |
| CoT-Diff$_{Qwen}$ | 48.3 | 49.1 | 59.2 | 61.4 | 58.3 | 58.2 |
| CoT-Diff$_{GPT-4o}$ | 53.2 | 53.3 | 66.0 | 67.8 | 62.9 | 62.7 |
| CoT-Diff$_{Gemini}$ | 54.9 | 55.2 | 66.4 | 69.0 | 64.2 | 64.5 |

Table 6: User Study.

| Method | 3D Layout Accuracy | Visual Quality |
|---|---|---|
| FLUX | 2.71 | 3.56 |
| RPG | 3.57 | 3.75 |
| EliGen | 3.65 | 3.81 |
| CoT-Diff (Ours) | **4.08** | **4.12** |

the highest scores across all categories, highlighting the complementary benefits of semantic layout, depth, and 3D layout optimization.

**Robustness across Different MLLMs.** We further evaluate CoT-Diff with several different MLLMs, including Qwen2.5-VL [2], GPT-4o [1] and Gemini 2.5 Pro, as detailed in Table 5. All tested MLLMs significantly improve spatial consistency over the FLUX baseline, but CoT-Diff when paired with Gemini 2.5 Pro consistently achieves the best results, particularly in handling front and behind relations. This result indicates that Gemini 2.5 Pro offers superior semantic planning capabilities for 3D layout generation, ultimately leading to more accurate spatial alignment.

**User Study.** We conducted a user study comparing CoT-Diff against four baselines using 200 complex prompts from 3DSceneBench. Six evaluators rated 3D Layout Accuracy and Visual Quality on a 1-5 scale. As shown in Table 6, CoT-Diff significantly outperformed all baselines, achieving the highest scores in both Layout Accuracy (4.08) and Visual Quality (4.12). This demonstrates our method's ability to enhance spatial control without the fidelity degradation.

### 4.4 3D Layout Consistency

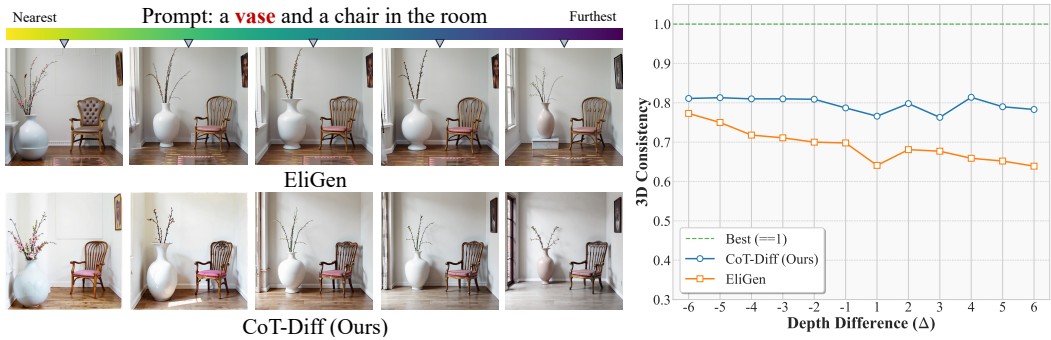

Figure 5: 3D consistency under depth variation. (Left) CoT-Diff accurately adjusts spatial placement across relative depths, while EliGen struggles to maintain correct object scale and distance. (Right) CoT-Diff consistently outperforms EliGen in 3D alignment across varying depth gaps.

This experiment evaluates how well different models maintain spatial consistency when object depth varies. We select two central objects from 3DSceneBench and construct a base 3D layout. One object is shifted along the camera axis by a multiple of $\Delta$, creating modified layouts. Each generated image is compared to the ground-truth layout by computing a depth-based consistency score. See Appendix H for details on setup and metric. Figure 5 presents both qualitative (left) and quantitative (right) results across depth differences. EliGen performs well at close range, but its consistency drops sharply as the depth difference increases. In contrast, CoT-Diff maintains stable consistency across all ranges, demonstrating stronger adherence to 3D layouts.

Table 7: Inference time and success rate of different methods.

| Method | Planning Time | Feedback Time | Generation Time | Whole Time | Success |
|---|---|---|---|---|---|
| FLux | - | - | 28.1 | 28.1 | 41.6 |
| RPG | 19.8 | - | 8.5 | 28.3 | 49.6 |
| EliGen | 5.2 | - | 55.7 | 60.9 | 53.2 |
| Image-CoT | - | - | 114.4 | 114.4 | 61.2 |
| CoT-Diff | 23.4 | 56.8 | 21.2 | 101.4 | 75.4 |

Table 8: Inference time across MLLM steps.

| MLLM step | Planning Time | Feedback Time | Generation Time | Whole Time |
|---|---|---|---|---|
| base(0) | 23.4 | - | 19.0 | 43.8 |
| 1 | 23.4 | 19.3 | 19.5 | 62.2 |
| 2 | 23.4 | 38.9 | 20.3 | 82.6 |
| 3 | 23.4 | 58.8 | 21.4 | 102.5 |
| 4 | 23.4 | 79.1 | 22.5 | 125.0 |
| 5 | 23.4 | 99.2 | 23.3 | 145.9 |
| ada (2.9) | 23.4 | 56.8 | 21.2 | 101.4 |

## 4.5 Runtime Efficiency and Spatial Success Rate

To evaluate the efficiency and effectiveness of CoT-Diff in complex scenes, we randomly sample 50 real prompts from 3DSceneBench and compare different methods in terms of runtime efficiency and spatial success rate. For efficiency, as shown in Table 7, while CoT-Diff's iterative feedback time adds overhead compared to simpler layout-based methods, its total inference time remains comparable to other CoT frameworks like Image-CoT [14]. This trade-off enables significantly improved spatial consistency. CoT-Diff achieves a 75.4% success rate, substantially outperforming baselines. In contrast, FLUX lacks layout modeling, while RPG and EliGen show limited spatial precision.

Furthermore, we evaluated the inference time across different MLLM optimization steps in Table 8. We found our adaptive early-stopping strategy (ada) achieves high performance in just 2.9 steps on average, striking an effective balance between generation quality and computational cost.

## 5 Conclusion

We propose CoT-Diff, a 3D-aware image generation framework that tightly couples MLLM reasoning with diffusion. It performs layout planning and generation jointly within a single diffusion round, leading to improved spatial structure and semantic alignment. By combining semantic and depth condition with a condition-aware attention mechanism, CoT-Diff enables precise and disentangled multimodal control. Considering the limmitations in Appendix K, we aim to integrate layout reasoning directly into the generative model for end-to-end CoT-lized generation in the future.

## Acknowledgement

This work was supported in part by the Natural Science Foundation of China (No. 62332002, 62202014, 62425101), and the Alibaba Innovative Research Program of Alibaba DAMO Academy.

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

# A  LLM Instruction for CoT-Diff

---

**Key Identity Parsing Prompt Template**

You are tasked with identifying and extracting all the real object names from a caption.

An object name refers to any tangible or physical entity mentioned in the caption.
Ensure not to include any adjectives or single-word descriptions that do not refer to a specific object, such as `"background."`

Please follow these instructions:

Identify all object names in the caption in the order they appear.
Maintain the exact wording of each object name as it is in the caption, including case consistency.
Output the object names in a Python list format.

For example, consider the following caption:    `<In-context Examples>`

Now, given the following caption, extract the object names in the same format:    `<caption>`

---

**MLLM-guided 3D Scene Planning Prompt Template**

As a 3D scene layout planner, generate a quantitative 3D layout (size, position) for specified entities based on a text caption.

Input:
1. A text caption describing the scene.
2. A list of important entity names in the scene.

Output: a JSON object with two keys:
`scene_parameters` and `entity_layout`.

  - `scene_parameters`: Describe the overall scene.

    - `scene_size` (meters): Approximate scale of the main subject area.

    - `camera_pitch_angle` (degrees): Vertical camera angle (positive = downward).

  - `entity_layout`: A list of objects, each including:

    - `entity_name`: Name of the entity.

    - `size`: [length, width, height] in meters. Should be large enough to be visible in the scene (each dimension $>$ `scene_size`/10).

    - `position`: [X, Y, Z] in meters, centered around the ground origin ($Y = 0$). Enforce explicit spatial relationships in the caption.

Coordinate System: Right-handed. Origin $(0, 0, 0)$ is the ground center. $+X =$ right, $+Y =$ up, $+Z =$ into the scene.

---

**Prompt Template: Iterative 3D Layout Optimization Assistant**

**System Role:** You are an AI Layout Optimization Assistant. Your core mission is to iteratively refine 3D JSON layouts through multi-turn dialogue with the user.

**Key Principles:**
1. **Entity Focus**: Evaluate and modify only the `entity_list` items for each turn.
2. **Viewer's Perspective**: Interpret all spatial terms (e.g., "left", "right") from the viewer of the `generated_image`.
3. **Iterative Learning**: Improve layout step-by-step based on prior adjustments.
4. **Adhere to Task Definition**: Strictly follow user-provided format and criteria.
5. **Historical Context**: Consider past actions and outcomes to inform new proposals.

**Task:**
Iteratively optimize the 3D JSON layout to align the `generated_image` with the `text_caption`, improving the clarity and spatial correctness of entities in the `entity_list`.

**Per-Iteration Inputs:**
- `text_caption`: (string) natural language description of the scene.
- `entity_list`: (list of strings) entities to optimize.
- `current_layout`: (JSON) 3D layout with size = [X_len, Z_width, Y_height] and position = [X, Y, Z], Y = 0 is ground.
- `generated_image`: the image rendered from the current layout.

**Step-by-Step Process:**
**Step 1: Parse Inputs**
Receive and acknowledge all inputs.

**Step 2: Evaluate Alignment (for `entity_list`)**
2.1 Discernibility: Is each entity clearly visible?
2.2 Verifiability: Are their described attributes verifiable?
2.3 Spatial Accuracy: Are spatial relations correct from viewer's perspective?
2.4 Determine `isaligned`: Set to true if 2.1–2.3 pass; else false.

**Step 3: Diagnose Misalignment (if `isaligned` = false)**
3.1 Identify which entities failed which checks.
3.2 Classify each as **Incorrect** or **Insufficient**.
3.3 Refer to previous adjustments and compare changes.

**Step 4: Revise Layout**
4.1 Strategize updates to `size`, `position`, or orientation of problematic entities.
4.2 Adjust other entities only if they cause conflicts.
4.3 Ensure layout format is valid.

**Step 5: Generate Output**
5.1 Text: Explain `isaligned` result and edits made.
5.2 JSON:
`{ "isaligned": <true/false>, "optimized_layout": <layout_object> }`

**User Prompt Format:**
```
text_caption: <caption>
entity_list: <entities>
current_layout: <layout>
generated_image: <image>
```

Table 9: Performance across different optimization steps. 'ada (2.9)' is our adaptive strategy.

| Steps | Basic Rela | | | | | | Hard Rela | |
|---|---|---|---|---|---|---|---|---|
| | Front | Behind | Front Left | Front Right | Back Left | Back Right | Multi Rela | Complex |
| base(0) | 51.3 | 51.2 | 62.2 | 61.4 | 59.9 | 56.7 | 41.7 | 48.5 |
| 1 | 53.3 | 51.9 | 63.4 | 66.0 | 61.2 | 59.8 | 45.8 | 48.4 |
| 2 | 57.6 | 53.6 | 66.6 | 67.1 | 62.8 | 63.3 | 47.9 | 50.4 |
| 3 | 55.2 | 54.5 | 67.5 | 69.5 | 62.2 | 64.9 | 47.6 | 49.7 |
| 4 | 56.0 | 54.5 | 68.3 | 69.9 | 61.9 | 64.9 | 48.8 | 51.4 |
| 5 | 56.2 | 56.4 | 67.9 | 71.2 | 64.4 | 66.1 | 51.0 | 51.8 |
| ada (2.9) | 54.9 | 55.3 | 66.4 | 69.0 | 64.1 | 64.5 | 48.7 | 50.8 |

Table 10: Comparison with Unified and Image CoT models.

| Steps | Basic Rela | | | | | | Hard Rela | |
|---|---|---|---|---|---|---|---|---|
| | Front | Behind | Front Left | Front Right | Back Left | Back Right | Multi Rela | Complex |
| Janus-Pro | 35.2 | 36.5 | 29.7 | 24.9 | 24.9 | 22.9 | 19.6 | 39.0 |
| VILA-U | 37.2 | 38.3 | 28.8 | 31.5 | 29.2 | 32.4 | 21.3 | 29.5 |
| Image-CoT | 48.2 | 49.7 | 56.1 | 61.2 | 56.7 | 57.6 | 39.7 | 46.6 |
| T2I-R1 | 47.0 | 46.8 | 47.2 | 45.3 | 50.4 | 51.1 | 35.5 | 41.5 |
| **CoT-Diff** | **54.9** | **55.2** | **66.4** | **69.0** | **64.2** | **64.5** | **48.7** | **50.8** |

## B  Training Details

We employed FLUX.1 dev as the pre-trained DiT. For each LoRA training, we utilize 8 A100 GPUs(80GB), a batch size of 1 per GPU. We employ the Prodigy optimizer [26] with safeguard warmup and bias correction enabled, setting the weight decay to 0.01 following OminiControl. For semantic LoRA $LoRA_{SL}$, we train the model 5000 iterations base on EliGen LoRA. For depth LoRA $LoRA_{DL}$, we train the model 30000 iterations.

## C  Performance across Different Optimization Steps

To validate our adaptive strategy, we evaluated the performance at different fixed optimization steps. As shown in Table 9, performance consistently improves as the number of steps increases, and typically saturates around 3 steps. Our adaptive method ('ada (2.9)') achieves considerable performance in just 2.9 steps on average, striking an effective balance between performance and efficiency.

## D  Comparison with Unified models and Image CoT models

We added a comparison with Janus-Pro [8], VILA-U [44], Image-CoT [14], and T2I-R1 [19]. The results, shown in Table 10, demonstrate CoT-Diff's superior performance in spatial reasoning and text-image alignment. This suggests that current unified generation-understanding models still lack explicit reasoning capabilities compared to our CoT-Diff framework. Furthermore, existing Image-CoT methods primarily implement CoT at the semantic or token level, whereas CoT-Diff introduces explicit 3D spatial reasoning, using an MLLM to generate and refine a physically interpretable 3D layout as precise guidance for the diffusion model.

## E  Details for Constructing 3DSceneBench

**Overview.** 3DSceneBench aims to evaluate the T2I model's compositional generation ability for complex spatial relationship prompts across basic- and hard-relations. For basic-relations prompts, we categorize non planar spatial relationships in the basic orientation in *six* cases: (1) front, (2) back, (3)

front left, (4) front right, (5) back left and (6) back right. These cases represent a mixture of horizontal and depth attributes defining the relative positions of typically two objects. The hard-relations prompts are designed to challenge models with increased compositional complexity, categorized into *two* main types based on how basic relations are combined or extended: (1)multi, combining two basic-relation prompts; (2)complex, expanding a basic-relation prompt into a complex scene description.

Table 11: Object Categories, Associated Scenes, and Possible Objects.

| Object Category | Associated Scenes | Possible Objects |
|---|---|---|
| **Animals** | in the desert, in the jungle, on the road, on the beach | dog, mouse, sheep, cat, cow, chicken, turtle, giraffe, pig, butterfly, horse, bird, rabbit, frog, fish |
| **Indoor** | in the room, in the studio, in the apartment, in the library | bed, desk, key, chair, vase, candle, cup, phone, computer, bowl, sofa, balloon, plate, refrigerator, bag, painting, suitcase, table, couch, clock, book, lamp, television |
| **Outdoor** | in the desert, on the street, on the road, on a snowy landscape | car, motorcycle, backpack, bench, train, airplane, bicycle |
| **Person** | All | woman, man, boy, girl |

**Complex prompt generation.** We choose 50 diverse objects from the MS-COCO dataset. These objects are categorized into four types based on their nature: indoor, outdoor, animal, and person. To provide contextual variation, four distinct scene contexts are defined for all categories except "person". Prompt generation is structured around basic spatial relationships and their compositions. For basic-relations prompts, we employed a template of the form "a {object1} {relation} a {object2} {scene}". Using this template with selected objects and defined scenes, we generated 200 prompts for each of the six basic-relation categories (front, back, front left, front right, back left, back right). Multi-relation prompts (200 in total) are created by combining pairs of the generated basic-relation prompts. Complex-relation prompts (200 in total) are generated by leveraging GPT-4o to expand selected basic-relation prompts into descriptions of more elaborate scenes. The final dataset undergo a human filtering process, similar to methods used in other benchmarks [28, 29], where human annotators review the generated prompts for suitability and quality before final inclusion.

# F   GPT Instruction for Calculating % Complex

To measure whether a prompt contains complex spatial relationships, following the approach of [29], we ask GPT4 with the yes or no binary question using the following instructions. " *You are an assistant to evaluate if the text prompt contains complex spatial relationships. Evaluate if complex spatial relationships are contained in the text prompt: PROMPT, The answer format should be YES or NO, without any reasoning.* ". Formally, the % complex of each test dataset $\mathcal{C}_{test}$ is calculated as $\%\text{Complex}(\mathcal{C}_{test}) = 1/|\mathcal{C}_{test}| \sum_{\mathbf{c} \in \mathcal{C}_{test}} \mathbb{1}(\text{GPT}_{complex}(\mathbf{c}) == \text{Yes})$, where $\text{GPT}_{complex}(\mathbf{c})$ is the binary answer of complex from GPT.

# G   Details for Evaluation

We adopt different evaluation protocols for each benchmark. For **3DSceneBench**, we use UniDet to evaluate spatial layout consistency. For **DVMP**, we ask GPT4o with a detailed score rubric [7], following the approach of [29]. For **T2I-CompBench**, we follow the official evaluation scripts released [17].

**GPT-based Evaluation.** For DVMP dataset, we leverage GPT-4o to evaluate the image-text alignment between the prompt and the generated image. The evaluation is based on a scoring scale from 1 to 5, where a score of 5 represents a perfect match between the text and the image, and a score of 1 indicates that the generated image completely fails to capture any aspect of the given prompt.

Table 12 presents the complete prompt with a full scoring rubric. We convert the original score scale $\{1, 2, 3, 4, 5\}$ to $\{0, 25, 50, 75, 100\}$, which is reflected in the reported results.

Table 12: Full LLM instruction for evaluation.

You are my assistant to evaluate the correspondence of the image to a given text prompt.
Focus on the objects in the image and their attributes (such as color, shape, texture), spatial layout, and action relationships. According to the image and your previous answer, evaluate how well the image aligns with the text prompt: **[PROMPT]**

Give a score from 0 to 5, according to the criteria:
5: image perfectly matches the content of the text prompt, with no discrepancies.
4: image portrayed most of the content of the text prompt but with minor discrepancies.
3: image depicted some elements in the text prompt, but ignored some key parts or details.
2: image depicted few elements in the text prompt, and ignored many key parts or details.
1: image failed to convey the full scope in the text prompt.

Provide your score and explanation (within 20 words) in the following format:
### SCORE: score
### EXPLANATION: explanation

**UniDet-based Spatial Relationship Evaluation.** For 3DSceneBench evaluation, we decompose a complex spatial relationship (e.g., "front left") into its constituent one-dimensional components: a horizontal relationship ("left" or "right") and a depth relationship ("front" or "back").

As a prerequisite, we utilize the UniDet [54] model to detect relevant objects in the generated image and obtain their bounding boxes and positional information.

Following detection, we evaluate the presence and accuracy of each individual decomposed relationship based on a metric similar to the UniDet-based approach described in T2I-CompBench [17].

For horizontal relationships ("left" or "right"), we compare the bounding box center coordinates of the involved objects and get a horizontal score. For depth relationships ("front" or "back"), we leverage depth information, typically obtained via depth estimation alongside detection and get a depth score.

The final score for a complex relationship is computed as the average of the evaluation scores of its constituent one-dimensional relationships.

# H 3D Consistency Metric

The 3D layout consistency experiment evaluates depth consistency under controlled 3D layout variations. A synthetic scene is configured with two objects selected from the 3DSceneBench, initially placed centrally. Layout modification involves shifting one object along the camera axis by a distance $d_1$, defined as a multiple of a predefined unit distance and sampled across 12 discrete values to cover a range of changes.

Following image generation for each modified layout, a monocular depth estimation model [46] is applied to the generated image to obtain depth information. From the resulting depth map, the depth difference $d_2$ between the centers of the two objects is measured. The 3D consistency score is then computed by comparing this measured depth difference $d_2$ to the original 3D shift distance $d_1$, using the following formula:

$$3D\ Consistency = 1 - \frac{|d_1 - d_2|}{|d_1| + |d_2|}$$

This score quantifies how well the depth relationship perceived in the generated image aligns with the intended depth change in the 3D layout, with higher values indicating better 3D consistency.

# I Further Visualization

Figure 6 displays images generated by CoT-Diff on 3DSceneBench. We randomly selected 8 prompts from 3DSceneBench and generated images using 5 different random seeds. Overall, the generated images show strong alignment with the input prompts while maintaining naturalness and high quality.

## J   More Visualization Results of challenging spatial scenarios

Figure 7 shows more visualization examples of challenging spatial scenarios. For each scenario, we display the 3D layout planned by CoT-Diff alongside five results generated using different random seeds, demonstrating the precise spatial control capabilities of our method.

## K   Limitations

Although CoT-Diff demonstrates strong spatial control in complex scenes, it has two primary limitations. First, the reliance on a MLLM for per-step reasoning introduces non-negligible inference cost. Second, layout planning is currently fully dependent on the MLLM, without any spatial reasoning capacity integrated into the generative model itself. Future work could explore endowing the generator with internal reasoning ability to enable more efficient and unified control.

## L   Boarder impacts

CoT-Diff enhances spatial alignment and composition fidelity in text-to-image generation, benefiting creative and educational applications. However, risks remain, including misuse for fake content and bias inherited from MLLMs. We encourage responsible deployment, bias auditing, and transparency in future development.

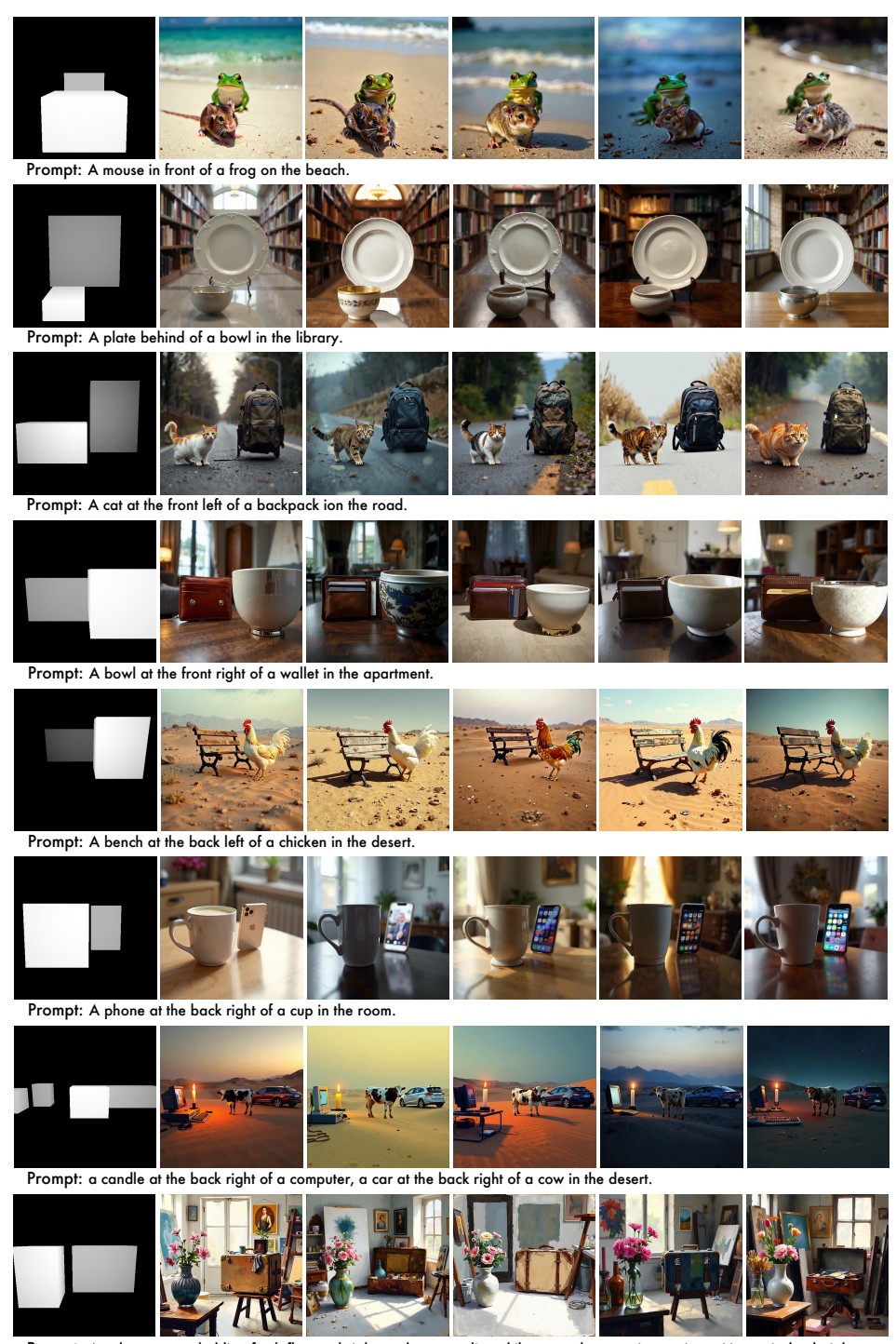

Prompt: A mouse in front of a frog on the beach.

Prompt: A plate behind of a bowl in the library.

Prompt: A cat at the front left of a backpack ion the road.

Prompt: A bowl at the front right of a wallet in the apartment.

Prompt: A bench at the back left of a chicken in the desert.

Prompt: A phone at the back right of a cup in the room.

Prompt: a candle at the back right of a computer, a car at the back right of a cow in the desert.

Prompt: An elegant vase holding fresh flowers brightens the art studio, while a travel-worn suitcase sits waiting to its back right.

Figure 6: CoT-Diff visualizations on 3DSceneBench: 3D layout depth maps (left) and corresponding generated images (right), from 8 random prompts with 5 random seeds each.

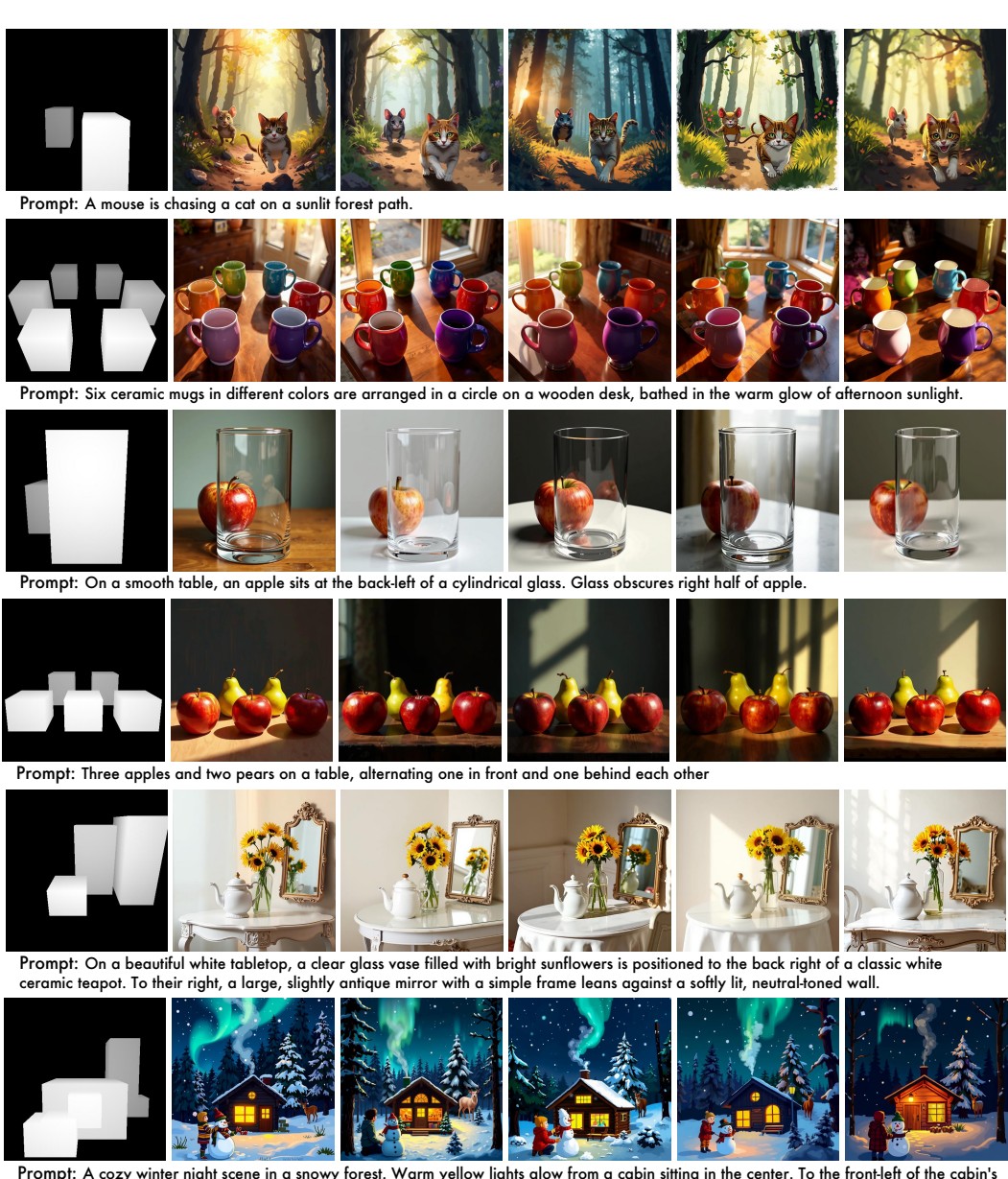

Prompt: A mouse is chasing a cat on a sunlit forest path.

Prompt: Six ceramic mugs in different colors are arranged in a circle on a wooden desk, bathed in the warm glow of afternoon sunlight.

Prompt: On a smooth table, an apple sits at the back-left of a cylindrical glass. Glass obscures right half of apple.

Prompt: Three apples and two pears on a table, alternating one in front and one behind each other

Prompt: On a beautiful white tabletop, a clear glass vase filled with bright sunflowers is positioned to the back right of a classic white ceramic teapot. To their right, a large, slightly antique mirror with a simple frame leans against a softly lit, neutral-toned wall.

Prompt: A cozy winter night scene in a snowy forest. Warm yellow lights glow from a cabin sitting in the center. To the front-left of the cabin's main entrance, a child is busy building a snowman under the falling snow. Near a large snow-covered pine tree to the back-right of the cabin, a deer stands watching quietly. Smoke gently rises from the chimney, and in the sky, the northern lights shimmer above the treetops.

Figure 7: More visualization results of challenging spatial scenarios.

