# OpenReview forum: "CoT-lized Diffusion: Let's Reinforce T2I Generation Step-by-step"
_NeurIPS.cc/2025/Conference — NeurIPS 2025 poster_

### Official Review · Reviewer_sj9U · 2025-06-28

**Clarity:** 3
**Significance:** 3
**Originality:** 3
**Rating:** 5
**Confidence:** 4

**Summary:**

This paper proposes CoT-lized Diffusion, a pipeline that improves spatial and semantic consistency in text-to-image generation by incorporating step-by-step layout reasoning. The core idea is to use GPT-4 to decompose a long or complex prompt into sub-goals, each guiding a 3D layout. A 3D scene planner integrates these sub-layouts into a unified structure, which is then fed into a conditional diffusion model (FLUX.1-schnell) to generate the final image. The method is motivated by the need to address ambiguities in natural language descriptions, especially those involving spatial relations. Experimental results show that the layout accuracy and fidelity to prompt details is better than the baselines.

**Questions:**

- Overclaiming CoT: Could the authors clarify in what way their method performs reasoning beyond spatial layout decomposition? Do you claim any general CoT behavior or just layout disambiguation?
- Backbone bias: Since FLUX.1-schnell is state-of-the-art, please clarify the choice of baseline. Have you tested the method on a weaker or standard backbone like SDXL?
- Qualitative correctness: In Fig. 4’s last column, the key action (“pointing”) is not present. Could you discuss failure cases or limitations of action grounding?

**Ethical Concerns:**

["NO or VERY MINOR ethics concerns only"]

**Final Justification:**

This paper presents a technically solid and well-motivated framework for step-by-step spatial reasoning in text-to-image diffusion models. The experiments are thorough and show consistent improvements over strong baselines. The authors have provided detailed clarifications in the rebuttal, addressing the main concerns. Overall, the contribution is significant and timely. I recommend acceptance.

**Limitations:**

Partially. The paper does not explicitly address limitations such as over-reliance on 3D layout accuracy, lack of end-to-end differentiability, or the societal risks of generation failures (e.g., hallucinated actions or biases in spatial understanding). A more complete discussion would improve transparency.

**Quality:**

3

**Strengths And Weaknesses:**

The main strengths are listed below.
- (1) The main issue of this paper aiming at is clear, i.e., prompt ambiguity, especially for spatial layouts.
- (2) The proposed modular architecture cleanly separates prompt decomposition, layout generation, and image synthesis by leveraging GPT-4 to structure prompts into layout subgoals is an adaptation of CoT-style reasoning to the vision domain.
- (3) This paper uses 3D layout as intermediate grounding which offer better controllability and interpretability.

However, several weaknesses are listed below.
- (1) The method addresses only spatial disambiguation rather than broader forms of reasoning (e.g., temporal, causal, or functional). The "CoT" framing may be misleading as it only affects layout.
- (2) This paper lacks of user study or human evaluation, despite the task involving semantics and perception. The evaluation relies solely on LLM scorer and visual examples.
- (3) Some details are missing in formulation. For instance, vt in Eq. 4 is undefined, which is important. Moreover, the loss optimization procedure is vaguely described.
- (4) FLUX.1-schnell is a powerful backbone. It is unclear whether the baselines use the same backbone or use others, e.g., SDXL. RPG is training-free, which can be applied to the same backbone.
- (5) Qualitative failure in core claim: As seen in Fig. 4 (last column), the image for “pointing at a distant sailboat” fails to depict the pointing action. This suggests the pipeline may not faithfully capture verb-level intent or fine-grained semantics.

---

> ### Author Rebuttal · Authors · 2025-07-30
>
> Thank you for your thoughtful and encouraging feedback. We have clarified the CoT scope, addressed technical details, and discussed limitations as suggested.
>
>
> ## Weakness
>
> > **Weakness 1**: The method addresses only spatial disambiguation rather than broader forms of reasoning (e.g., temporal, causal, or functional). The "CoT" framing may be misleading as it only affects layout.
>
> **Answer:**
> Thank you for your insightful comment. We framed our method as "CoT-lized" because it adopts the step-by-step reasoning paradigm from LLMs, using an iterative "plan-evaluate-refine" loop to guide generation. This describes the process of our method, not just its application domain.
> Furthermore, our framework can indeed handle broader inferential tasks, leveraging the MLLM's knowledge. For instance, given the prompt "Einstein's favorite musical instrument", our MLLM reasons that it is a violin and rewrites the prompt into a generatable form like "a beautiful violin." This demonstrating its capacity for factual and conceptual reasoning. We will clarify this point in our revised manuscript. Thank you for your valuable feedback.
>
> > **Weakness 2**: This paper lacks of user study or human evaluation, despite the task involving semantics and perception. The evaluation relies solely on LLM scorer and visual examples.
>
> **Answer:**
> Thank you for the valuable suggestion. In response, we have conducted a human evaluation to assess our method's semantic and perceptual performance. We asked six human evaluators to rate images generated for 200 complex prompts from 3DSceneBench. They scored our method (CoT-Diff) against four key baselines on a 1-5 scale across two criteria: 3D Layout Accuracy and Visual Quality.
>
> | Method        | 3D Layout Accuracy | Visual Quality |
> |---------------|--------------------|----------------|
> | FLUX          | 2.71               | 3.56           |
> | RPG           | 3.57               | 3.75           |
> | EliGen        | 3.65               | 3.81           |
> | Image-CoT     | 3.83               | 2.95           |
> | CoT-Diff (Ours) | 4.08               | 4.12           |
>
> As shown in Table, CoT-Diff significantly outperforming all baselines. Notably, our method achieves the highest scores in both Layout Accuracy (4.08) and Visual Quality (4.12), demonstrating a unique ability to enhance complex spatial control without the degradation in image fidelity seen in methods like Image-CoT.
>
> > **Weakness 3**: Some details are missing in formulation. For instance, vt in Eq. 4 is undefined, which is important. Moreover, the loss optimization procedure is vaguely described.
>
> **Answer:**
> We thank the reviewer for pointing out these omissions and apologize for the lack of clarity.
>
> To address your first point, **v_t** in Equation (4) represents **the velocity field predicted by our FLUX model at timestep t**. Formally, $v_t = v_Θ(z_t, t, C)$, where $C$ denotes the combined conditional inputs (e.g., depth map and text embeddings) and $Θ$ represents the model's parameters.
>
> Regarding the loss optimization procedure, we train our two LoRA modules (Semantic and Depth) independently using a **flow-matching loss**. The specific loss functions are as follows:
>
> 1. **For the Semantic LoRA module (L_SL):**
> $L_{SL} = E_{t,z,\epsilon}[||v_{\theta}(z, t, P, P_1, ..., P_k) - u_t(z|\epsilon)||^2]. $
> Here, $v_θ$ is the velocity field predicted by the Semantic LoRA-augmented model, $P, P1, ..., Pk$ are the global and local prompt embeddings, and $u_t(z|ε)$ is the target velocity field.
> 2. **For the Depth LoRA module (L_DM):**
> $L_{DM} = E_{t,z,\epsilon}[||v_{\Phi}(z, t, C_D, P_{emb}) - u_t(z|\epsilon)||^2]. $
> Here, $v_Φ$ is the velocity field predicted by the Depth LoRA-augmented model, $C_D$ is the depth condition, and $P_emb$ is the prompt embedding.
>
> We will update Section 3 with these precise formulations to improve clarity and reproducibility.
>
> > **Weakness 4**: FLUX.1-schnell is a powerful backbone. It is unclear whether the baselines use the same backbone or use others, e.g., SDXL. RPG is training-free, which can be applied to the same backbone.
>
> **Answer:**
> Thank you for this crucial point on experimental fairness.
>
> You are correct. In our initial experiments, we aimed to showcase each method at its best by using their officially recommended backbones (e.g., RPG with SDXL, EliGen with FLUX.1-dev). However, for the most direct comparison, **we highlight our results against EliGen in Tables 2 and 3 of our main text, a training-based method that also utilizes a FLUX backbone**. As shown in Tables 2 and 3 of our main text, CoT-Diff already demonstrates a significant performance advantage. This provides compelling evidence that our framework's superiority will hold when compared against **a training-free method like RPG** on the same architecture.
>
> For the sake of complete academic rigor, we agree that an explicit experiment on the FLUX.1-schnell backbone is essential for isolating the specific advantages of our CoT-Diff framework. Unfortunately, adapting RPG from its native SDXL (U-Net architecture) to FLUX (DiT architecture) is a non-trivial engineering task due to significant architectural disparities. This adaptation is already in progress, but its complexity prevented us from finalizing the results within the brief rebuttal period.
>
> Therefore, we commit to including this crucial experiment in the camera-ready version of the paper. We will adapt RPG to the FLUX.1-schnell backbone for a direct and fair comparison. This will further solidify the superiority of our CoT-Diff framework.
>
> > **Weakness 5**: Qualitative failure in core claim: As seen in Fig. 4 (last column), the image for “pointing at a distant sailboat” fails to depict the pointing action. This suggests the pipeline may not faithfully capture verb-level intent or fine-grained semantics.
>
> **Answer:**
> We acknowledge the limitation shown in this example. This is likely due to two factors:
> 1) Coarse Control Granularity: our method emphasizes controlling the 3D layout of full objects, but the hand is only a partial component of the human body. Guiding such fine-grained regions (e.g., hand gestures) via whole-body 3D bounding boxes is inherently difficult, and precise control at this level remains a challenging open problem;
> 2) Backbone Limitations: the backbone itself has limited capacity for faithfully capturing verb-level semantics—none of the compared methods successfully rendered the pointing action, indicating a general limitation in current diffusion models for action grounding.
>
> ## Question
>
> > **Question 1**:
> Overclaiming CoT: Could the authors clarify in what way their method performs reasoning beyond spatial layout decomposition? Do you claim any general CoT behavior or just layout disambiguation?
>
> **Answer:**
> As stated in W1, our main claim is the CoT-style generation paradigm, where MLLMs guide the diffusion process step by step, rather than claiming general-purpose CoT reasoning.
>
>
>
> > **Question 2**:
> Backbone bias: Have you tested the method on a weaker or standard backbone like SDXL?
>
> **Answer:**
> Thank you for this question regarding the generalizability.
>
> Our initial experiments used FLUX.1-schnell to show our framework's maximum potential. We agree that testing on SDXL is crucial for demonstrating generalizability.
>
> However, implementing and training our entire framework on SDXL is a significantly more demanding task than adapting a training-free method like RPG (as discussed in our response to W4). This training process is already underway, but given the substantial computational resources and time required, we were unable to obtain fully converged results and complete the evaluation within the brief rebuttal period.
>
> We note that prior work (e.g., MIGC [1]) has successfully applied semantic and 2D layout control to the SDXL backbone. This suggests that our 3D-aware approach would also be effective on SDXL. We commit to including this analysis in a future version of our paper.
>
>
>
> > **Question 3**:
> Qualitative correctness: Could you discuss failure cases or limitations of action grounding?
>
> **Answer:**
> As stated in W5, our pipeline faces challenges in fine-grained action grounding such as "pointing," due to both the coarse 3D control granularity and limitations of the diffusion backbone.
>
>
>
> ## Limitation
>
> > **Limitation 1**:
> The paper does not explicitly address limitations like over-reliance on 3D layout accuracy, lack of end-to-end differentiability, or societal risks from generation failures (e.g., hallucinations or spatial understanding biases).
>
> **Answer:**
> Indeed, our method exhibits a degree of reliance on the accuracy of upstream 3D layouts. If these input 3D layout representations themselves contain errors or lack robustness, the quality and spatial precision of the final generated images may be impacted, even with CoT-Diff's effective utilization. Furthermore, the iterative reasoning process of the MLLM in the current framework is not end-to-end differentiable, which limits the potential for large-scale holistic optimization and deeper learning. More broadly, as the reviewer mentioned, all large generative models face risks of hallucination and bias, and our model is no exception. This could include generating "hallucinated" content inconsistent with semantics, or exhibiting biases in spatial understanding derived from training data.
>
> However, CoT-Diff's strength lies in its explicit 3D layouts and iterative MLLM refinement, allowing for improved spatial understanding and self-correction during generation, thus mitigating spatially related hallucinations and error propagation. Future work will focus on improving the robustness of the 3D layout generation, exploring differentiable reasoning frameworks, and actively addressing biases and societal risks in models.
>
> [1] Zhou, Dewei, et al. "Migc: Multi-instance generation controller for text-to-image synthesis." Proceedings of the IEEE/CVF conference on computer vision and pattern recognition. 2024.

---

> > ### Comment · Reviewer_sj9U · 2025-08-04
> > **Rebuttal Feedback**
> >
> > Thank you to the authors for the substantial effort in addressing the reviewer comments and for providing additional experiments. I appreciate the technical depth and the willingness to clarify points regarding both implementation and evaluation. However, I still have two main concerns:
> > - Scope of “CoT-lized” in Title and Claims
> > The current title and framing (“CoT-lized Diffusion: Let's Reinforce T2I Generation Step-by-step”) strongly suggest that the CoT-style generation paradigm can address a broad spectrum of challenges in T2I, or even general reasoning tasks. However, the core experiments and the methodology all focus on spatial/layout disambiguation. While the “Einstein’s favorite musical instrument” example is a nice illustration, it is not clearly established (either in the main experiments or in ablations) that genuine CoT reasoning beyond spatial layout is being performed. For instance, I found that even direct prompts like “Einstein’s favorite musical instrument” generate violins in standard SD pipelines. If the framework truly supports broader reasoning abilities, there should be dedicated experiments to support this claim, rather than relying on isolated anecdotal cases.
> >
> > - Qualitative Failure Analysis and Fig. 4
> > The case in Fig. 4 (last column) is not itself a problem—failure cases are valuable—but the manuscript does not discuss or analyze them. The experimental narrative throughout the paper focuses on wins: every visual case presented is described as an example where the proposed method outperforms baselines. However, in this specific example, EliGen actually produces a more correct result (the pointing gesture), while CoT-Diff does not. The lack of discussion or analysis of such cases is problematic. Does this example suggest a fundamental limitation or trade-off of the CoT-Diff approach for handling complex prompts or fine-grained actions? More critical self-analysis would strengthen the paper and improve reader understanding of where the proposed method excels or falls short.
> >
> >
> > Overall, while I appreciate the additional clarifications and new experiments, I hope the authors can further clarify:
> > - Whether “CoT-lized” should be interpreted as generalizable beyond layout reasoning, and if so, it is desirable to provide experiments to demonstrate this. Otherwise a precise title may be good to attract correct audience.
> > - Include an honest, explicit discussion in the manuscript about cases where the proposed approach underperforms, such as the Fig. 4 failure, and analyze the underlying causes or trade-offs.

---

> > > ### Author Response · Authors · 2025-08-05
> > > **Response to Reviewer sj9U**
> > >
> > > Dear Reviewer sj9U,
> > > Thank you for your valuable follow-up and for highlighting the need for a more critical self-analysis.
> > >
> > > > **Feedback 1**:
> > > Whether “CoT-lized” should be interpreted as generalizable beyond layout reasoning, and if so, it is desirable to provide experiments to demonstrate this. Otherwise a precise title may be good to attract correct audience.
> > >
> > > **Answer:**
> > > We agree that our title suggested a broader reasoning scope than our experiments demonstrated. Your feedback is very helpful in clarifying this.
> > > To better reflect our focus on spatial reasoning and disambiguation, we will revise the title to:
> > > “CoT-Spatialized Diffusion: Let's Reinforce T2I Generation Step-by-step”
> > > We believe this new title is more precise and better aligns with our core contribution.
> > >
> > > > **Feedback 2**:
> > > Include an honest, explicit discussion in the manuscript about cases where the proposed approach underperforms, such as the Fig. 4 failure, and analyze the underlying causes or trade-offs.
> > >
> > > **Answer:**
> > > We agree that analyzing failure cases, such as the one in Fig. 4, is crucial for understanding our method's trade-offs.
> > >
> > > The Fig. 4 example ("pointing at a distant sailboat") reveals a key trade-off of our approach: while it excels at achieving correct spatial composition, it has difficulty rendering fine-grained attributions. For context, while a baseline like EliGen sometimes rendered the "pointing gesture," it largely failed the prompt's main spatial constraint—that the woman be "front right." In contrast, CoT-Diff reliably achieves this compositional accuracy, demonstrating its primary strength.
> > >
> > > To quantify this, we generated 20 images for this prompt with both methods across different random seeds:
> > >
> > > * CoT-Diff: Achieved the correct spatial layout in 80% (16/20) of images, with the gesture present in 40% (8/20).
> > >
> > > * EliGen: Produced the gesture in 65% (13/20) of images, but satisfied the spatial layout in only 25% (5/20).
> > >
> > > This trade-off stems directly from our design. CoT-Diff uses 3D bboxes for layout and entity-level control. This approach is highly effective for the global scene structure, but it lacks a mechanism for fine-grained, intra-object details, while the strong layout conditions may simultaneously weaken the influence of these subtle attributes from the text prompt. In the Fig. 4 example, our method controls the person's overall 3D position but lacks a direct way to guide a specific part of the body, like the hand, to form the pointing gesture. We acknowledge this as a key trade-off of our current approach.
> > >
> > > For future work, we plan to explore incorporating finer-grained control, potentially via prompt-level CoT (e.g., T2I-R1[1]) and reinforcement learning. Specifically, we envision a semantic-level CoT to understand, decompose, and refine the prompt, enhancing the influence of fine-grained attributes. This structured reasoning process could then be optimized using algorithms such as GRPO. We will add this analysis to the limitations section, as we believe this transparent discussion strengthens the paper.
> > >
> > > [1] Jiang, Dongzhi, et al. "T2i-r1: Reinforcing image generation with collaborative semantic-level and token-level cot." arXiv preprint arXiv:2505.00703 (2025).

---

> > > > ### Comment · Reviewer_sj9U · 2025-08-06
> > > > **No Further Questions**
> > > >
> > > > I appreciate the thorough responses. Looking forward to seeing the final version.

---

> > > > > ### Author Response · Authors · 2025-08-06
> > > > >
> > > > > Thank you very much for your positive feedback. We are glad our response addressed your concerns.

---

> ### Author Response · Authors · 2025-08-02
> **Updated Response to W4:  Fair Comparison on the FLUX**
>
> Dear Reviewer sj9U,
>
> We sincerely thank for your insightful suggestion (Weakness 4) regarding a direct comparison on the same backbone architecture. And we have conducted a new experiment to address this.
>
> Initially, we committed to adapting RPG to the FLUX backbone. During this process, we discovered that porting RPG from its native U-Net (in SDXL) to the DiT architecture (in FLUX) is non-trivial and results in a significant performance drop. This is likely because RPG's guidance mechanism is highly optimized for U-Net's specific structure.
>
> Therefore, to provide a more rigorous and truly fair evaluation, we pivoted to compare our CoT-Diff against a much stronger and more relevant baseline: RAG-Diffusion [1]. This recently accepted work (ICCV 2025) presents a state-of-the-art training-free method, conceptually similar to RPG, but is explicitly designed for and demonstrates excellent performance on the FLUX.1-dev backbone.
>
> The results are as follows:
>
> | Method | Backbone | Front | Behind | Front Left | Front Right | Back Left | Back Right | multi | complex |
> | --- | --- | --- | --- | --- | --- | --- | --- | --- | --- |
> | RPG | SDXL | 40.5 | 38.2 | 37.1 | 39.8 | 38.8 | 42.1 | 25.2 | 36.5 |
> | RAG-Diffusion | FLUX.1-dev | 40.8 | 43.8 | 42.2 | 44.2 | 41.7 | 48.6 | 30.1 | 37.0 |
> | **CoT-Diff** | FLUX.1-schnell | **54.9** | **55.2** | **66.4** | **69.0** | **64.2** | **64.5** | **48.7** | **50.8** |
>
> As shown, these results offer conclusive evidence that the performance gains originate from our novel CoT-Diff framework—not merely the backbone—by decisively outperforming a more formidable and relevant baseline on the same FLUX architecture, thus fully addressing the reviewer's concern. We will incorporate these findings into our revised manuscript.

---

### Official Review · Reviewer_QBNx · 2025-06-30

**Clarity:** 3
**Significance:** 3
**Originality:** 3
**Rating:** 5
**Confidence:** 4

**Summary:**

The key contribution of this paper is the application of Chain-of-Thought (CoT) abilities from large multimodal models to Text-to-Image (T2I) tasks, which improves spatial layout and 3D perception. To assess these improvements, the authors have designed the 3DSceneBench benchmark for evaluating structure-based reasoning. The paper presents a strong motivation for this research.

**Questions:**

1.	Is the design of some prompts problematic? For example, in Figure 4 with the prompt, "a car behind a bench on a snowy landscape," the spatial structure is correct in the results from both CoT-Diff and EliGen. The perceived correctness in both is due to different viewing angles, as the prompt only specifies the front-behind relationship without left-right positioning. How should such cases be judged during the actual labeling process, and could this ambiguity affect the other evaluation metrics?
2.	The authors do not state the resolution of the generated images. Could the image size have an impact on the results?

**Ethical Concerns:**

["NO or VERY MINOR ethics concerns only"]

**Final Justification:**

After considering the rebuttal and discussions, I believe the authors have made a strong effort to address the key concerns, and my overall impression of the paper has improved.

**Limitations:**

1.	The evaluation section of this paper is somewhat insufficient.
a)	 In the "Main Results of CoT-Diff" section, the authors state that traditional diffusion models perform poorly in spatial tasks due to a lack of layout guidance. However, the LC method, which incorporates layout guidance, shows relatively the worst performance in the metrics. The authors do not provide a more detailed analysis of this counterintuitive result.
b)	 In the results analysis, the authors generally describe performance improvements but do not offer a detailed breakdown of individual results. For instance, in the "Ablation Study of CoT-Diff," the performance on the "Front" category decreases after adding "+Semantic Layout." The authors should briefly explain the potential reasons for this decline.
c)	An evaluation of the number of inference steps is missing. The authors have not analyzed how many steps are typically required for the model to generate images with well-composed spatial structures. As a major limitation cited in the paper is the high inference cost, such an evaluation is crucial.
d)	There have been other works this year that have also applied CoT to image generation (as listed below). The authors should cite or compare their work against these papers in the final version, clarifying the key differences and advantages of their proposed method.
[1] Guo, Ziyu, et al. "Can We Generate Images with CoT? Let's Verify and Reinforce Image Generation Step by Step." arXiv preprint arXiv:2501.13926 (2025).
[2] Jiang, Dongzhi, et al. "T2i-r1: Reinforcing image generation with collaborative semantic-level and token-level cot." arXiv preprint arXiv:2505.00703 (2025).

**Paper Formatting Concerns:**

This paper is well written and there are no concerns about the writing format.

**Quality:**

3

**Strengths And Weaknesses:**

Pros:
1.	This work is well-motivated and novel. The understanding of 3D structures in images is crucial for tasks such as image generation and content comprehension. Furthermore, the paper introduces an innovative, CoT-based approach to enhance the model's understanding of spatial structures in T2I tasks.
2.	The contributions are substantial. In addition to proposing the Cot-Diff model, the authors have also designed a sound evaluation framework, 3DSceneBench.
3.	The paper is concisely written and easy to follow, clearly articulating the primary work presented.

Weaknesses:

1. The proposed CoT-Diff framework requires invoking a large MLLM (GPT-4o or Gemini) at each denoising step, resulting in substantial inference cost. Although the authors mention this in the appendix and describe inference time as “reasonable,” the iterative nature of the reasoning process raises concerns regarding scalability and real-world applicability.
2. While the method is framed as a CoT approach, its reasoning capabilities are primarily restricted to spatial layout understanding. Broader reasoning capabilities (temporal, causal, or functional reasoning) are not explored.
3. Errors in early stages (misinterpretation of spatial prompts or poor initial layout) could propagate and negatively affect the final output. The paper does not sufficiently analyze failure modes or the model’s ability to recover from such errors.

---

> ### Author Rebuttal · Authors · 2025-07-30
>
> We sincerely thank you for your recognition of our work. Based on your suggestions, we have added further analysis, clarified key experimental details, and included comparisons with recent CoT-based methods to strengthen the paper.
>
>
>
> ## Question
>
> > **Question 1**:
> Is the design of some prompts problematic? For example, in Figure 4 with the prompt, "a car behind a bench on a snowy landscape," the spatial structure is correct in the results from both CoT-Diff and EliGen. The perceived correctness in both is due to different viewing angles, as the prompt only specifies the front-behind relationship without left-right positioning. How should such cases be judged during the actual labeling process, and could this ambiguity affect the other evaluation metrics?
>
> **Answer:**
> Thank you for raising this question. In our evaluation, spatial relations such as "behind" are defined with respect to the camera (observer) viewpoint, which provides a consistent and objective reference frame. Without this convention, viewpoint rotation could arbitrarily flip spatial relations (e.g., “behind” becoming “front” or “right”), making evaluation unreliable. This camera-centric assumption is also adopted by established benchmarks such as T2I-CompBench[1].
>
>
>
> > **Question 2**:
> Could the image size have an impact on the results?
>
> **Answer:**
> Our current implementation generates images at a resolution of 512×512. Since our method relies on accurate 3D layout modeling, low-resolution settings (e.g., 128×128) may introduce errors in rendered depth, leading to hallucinations. However, we observe that at 512×512 or higher resolutions, the depth maps are sufficiently accurate to guide high-quality image generation.
>
>
> ## Limitation
>
> > **Limitation 1 a)**:
> The LC method incorporates layout guidance, yet it performs worse than traditional diffusion models in the reported metrics. A more detailed analysis is needed to explain this counterintuitive result.
>
> **Answer:**
> The seemingly counterintuitive performance of LC arises from several limitations in its design:
>
> 1) Backbone Choice: LC uses Stable Diffusion 1.5 as its backbone, which has relatively limited image quality and spatial understanding capabilities compared to more advanced models (e.g., FLUX). This constrains the upper bound of its spatial reasoning performance.
>
> 2) Conditioning Mechanism: Although LC accepts 3D layout input, it does so through a sparse-depth LoRA module trained on top of a ControlNet pretrained for dense depth. The sparse layout signal is thus weakly injected into the model through an indirect path (LoRA → ControlNet → SD), which reduces its effectiveness in enforcing correct spatial structure. Moreover, this training approach is also known to be challenging to converge and can even degrade the backbone's inherent generative capabilities.
>
> 3) Lack of Semantic Disentanglement: LC lacks a semantic module like the Semantic LoRA in CoT-Diff, making it more prone to object confusion when generating scenes with multiple entities. For example, in Fig. 4 (column 2), both the car and the bench are generated as benches, failing to represent the intended spatial relationship (“car behind a bench”).
>
> Together, these factors explain why LC, despite incorporating layout input, underperforms in spatially grounded generation tasks.
>
> > **Limitation 1 b)**:
> The results analysis lacks fine-grained interpretation. For example, in the ablation study, the "Front" category shows a performance drop after adding the "+Semantic Layout" module. The authors should briefly explain this decline.
>
> **Answer:**
> Thank you for your meticulous review of our ablation study. You have correctly identified the slight performance dip in the "Front" category after adding "+Semantic Layout," which is an excellent question. We believe this phenomenon is not an anomaly and can be attributed to the following factors:
>
> 1) Conflict between Semantic and Geometric Cues: The Semantic Layout module injects entity-level semantic information into specific image regions (as described in lines 143–144). However, in the absence of corresponding depth maps to enforce geometric structure, this strong semantic bias may conflict with the model’s pre-existing handling of simple spatial relations such as “front.” This can temporarily interfere with the model’s generation patterns and lead to slight performance drops.
>
> 2) Importance of Depth Information: This decline further confirms the necessity and effectiveness of adding the Depth Map module. As shown in Table 4, once depth information is introduced on top of the semantic layout, the model not only resolves occlusions and object ordering issues but also significantly improves performance in the “Front” category (from 44.5 to 51.3). This highlights the complementary nature of semantic and geometric guidance: while semantics guide content, depth provides spatial anchoring in 3D space.
>
> 3) The characteristics of the dataset: Our training data is constructed based on annotations from EliGen, and a similar pattern was observed in EliGen's results (as seen in Table 2 of our main text), where front/behind performance also dipped when only semantic layouts were introduced. Therefore, the decline we observe may be partially inherited from the limitations of the source dataset.
>
> > **Limitation 1 c)**:
> The paper lacks an analysis of how many inference steps are needed to produce spatially coherent images, which is important given that high inference cost is a stated limitation.
>
> **Answer:**
> We evaluates the performance with a fixed number of MLLM optimization steps at varying total steps. It can be observed that as the number of steps increases, the system's performance progressively improves, tending to converge after 3 steps.
>
> Accordingly, we designed an adaptive early-stopping strategy: once the model determines the layout is correct, it ceases further evaluation, with a maximum of 5 evaluation steps. This strategy balances performance and time overhead, achieving considerable performance with an average of 2.9 steps.
>
> | Step         | Front | Behind | Front Left | Front Right | Back Left | Back Right | Multi | Complex |
> |--------------|-------|--------|------------|-------------|-----------|------------|-------|---------|
> | base(0)      | 51.3  | 51.2   | 62.2       | 61.4        | 59.9      | 56.7       | 41.7  | 48.5    |
> | 1            | 53.3  | 51.9   | 63.4       | 66.0        | 61.2      | 59.8       | 45.8  | 48.4    |
> | 2            | 57.6  | 53.6   | 66.6       | 67.1        | 62.8      | 63.3       | 47.9  | 50.4    |
> | 3            | 55.2  | 54.5   | 67.5       | 69.5        | 62.2      | 64.9       | 47.6  | 49.7    |
> | 4            | 56.0  | 54.5   | 68.3       | 69.9        | 61.9      | 64.9       | 48.8  | 51.4    |
> | 5            | 56.2  | 56.4   | 67.9       | 71.2        | 64.4      | 66.1       | 51.0  | 51.8    |
> | ada (2.9)    | 54.9  | 55.3   | 66.4       | 69.0        | 64.1      | 64.5       | 48.7  | 50.8    |
>
>
> > **Limitation 1 d)**:
> Recent works have also explored applying CoT to image generation. The paper should cite and compare with these methods, clarifying the key differences and advantages of the proposed approach.
>
> **Answer:**
> Thank you for bringing these relevant recent works to our attention. We have studied them and added experimental comparisons.
>
> Key Differences and Advantages:
> The core difference between our CoT-Diff and these methods lies in the fundamental mechanism of the "Chain-of-Thought." Other methods (e.g., Image-CoT[2], T2I-R1[3]) implement CoT at the semantic or token level, refining text prompts or internal features iteratively. In contrast, CoT-Diff introduces explicit 3D spatial reasoning, using an MLLM to generate and refine a physically interpretable 3D layout, which serves as precise guidance for the diffusion model. This structured approach excels at handling complex spatial relationships and ambiguities.
>
> Experimental Comparison:
> We evaluated these methods on our 3DSceneBench benchmark. The results clearly show CoT-Diff's superior performance in spatial reasoning, particularly in challenging categories like "Front/Back Left/Right," "Multi Relation," and "Complex."
>
> We will include these works in our "Related Work" section and add the comparison to the "Experiments" section in the final version to demonstrate the novelty and effectiveness of our method.
>
> | Method      | Front | Behind | Front Left | Front Right | Back Left | Back Right | multi | complex |
> | :---------- | :---- | :----- | :--------- | :---------- | :-------- | :--------- | :---- | :------ |
> | janus-pro   | 35.2  | 36.5   | 29.7       | 24.9        | 24.9      | 22.9       | 19.6  | 39.0    |
> | vila-u      | 37.2  | 38.3   | 28.8       | 31.5        | 29.2      | 32.4       | 21.3  | 29.5    |
> | Image-CoT | 48.2  | 49.7   | 56.1       | 61.2        | 56.7      | 57.6       | 39.7  | 46.6    |
> | t2i-r1      | 47.0  | 46.8   | 47.2       | 45.3        | 50.4      | 51.1       | 35.5  | 41.5    |
> | **CoT-Diff**    | **54.9**  | **55.2**   | **66.4**       | **69.0**        | **64.2**      | **64.5**       | **48.7**  | **50.8**    |
>
> [1] Huang, Kaiyi, et al. "T2i-compbench: A comprehensive benchmark for open-world compositional text-to-image generation." Advances in Neural Information Processing Systems 36 (2023): 78723-78747.
>
> [2] Guo, Ziyu, et al. "Can We Generate Images with CoT? Let's Verify and Reinforce Image Generation Step by Step." arXiv preprint arXiv:2501.13926 (2025).
>
> [3] Jiang, Dongzhi, et al. "T2i-r1: Reinforcing image generation with collaborative semantic-level and token-level cot." arXiv preprint arXiv:2505.00703 (2025).

---

> > ### Comment · Reviewer_QBNx · 2025-08-06
> >
> > I have read through the authors’ response and appreciate the effort put into addressing the reviewers’ comments. The rebuttal is thorough and well-organized, with clear and specific answers to each concern. I sincerely apologize to the authors for not including a more thorough analysis of the paper’s weaknesses in my original review.
> >
> >
> >
> > Weaknesses:
> > 1. The proposed CoT-Diff framework requires invoking a large MLLM (GPT-4o or Gemini) at each denoising step, resulting in substantial inference cost. Although the authors mention this in the appendix and describe inference time as “reasonable,” the iterative nature of the reasoning process raises concerns regarding scalability and real-world applicability.
> > 2. While the method is framed as a CoT approach, its reasoning capabilities are primarily restricted to spatial layout understanding. Broader reasoning capabilities (temporal, causal, or functional reasoning) are not explored.
> > 3. Errors in early stages (misinterpretation of spatial prompts or poor initial layout) could propagate and negatively affect the final output. The paper does not sufficiently analyze failure modes or the model’s ability to recover from such errors.

---

> > > ### Author Response · Authors · 2025-08-06
> > > **Response to Reviewer QBNx**
> > >
> > > Dear Reviewer QBNx,
> > >
> > > We sincerely thank you for your thoughtful feedback and analysis of our work. We will answer each of the identified weaknesses below.
> > >
> > > > **Weakness 1**: The proposed CoT-Diff framework requires invoking a large MLLM (GPT-4o or Gemini) at each denoising step, resulting in substantial inference cost. Although the authors mention this in the appendix and describe inference time as “reasonable,” the iterative nature of the reasoning process raises concerns regarding scalability and real-world applicability.
> > >
> > > **Answer:**
> > > We thank the reviewer for concerns regarding computational overhead.
> > >
> > > CoT-Diff's MLLM is not invoked at every denoising step. Its primary cost is concentrated in the initial "thinking" phase for layout refinement.
> > >
> > > Inspired by this, we designed an adaptive early-stopping strategy: the MLLM ceases evaluation once the layout is deemed correct (max 5 steps), averaging just 2.9 steps. This effectively balances performance and efficiency, as shown in the inference time cost table in Reviewer XeQc **W1**.
> > >
> > > What's more, we have benchmarked CoT-Diff against several baseline methods. Please refer to the method time table in Reviewer XeQc **Q1**. Our method keeps inference cost manageable and competitive, especially when considering the significant leap in generation quality.
> > >
> > > We argue this front-loaded computational cost is a reasonable trade-off for the significant improvements in generation quality and spatial accuracy. CoT-Diff's value in layout-critical applications like game asset creation, virtual staging, and film pre-visualization justifies this expense.
> > >
> > > > **Weakness 2**: While the method is framed as a CoT approach, its reasoning capabilities are primarily restricted to spatial layout understanding. Broader reasoning capabilities (temporal, causal, or functional reasoning) are not explored.
> > >
> > > **Answer:**
> > > Thank you for your insightful comment.
> > >
> > > We framed our method as "CoT-lized" because it adopts the step-by-step reasoning paradigm from LLMs, using an iterative "plan-evaluate-refine" loop to guide generation. This describes the process of our method, not just its application domain.
> > >
> > > Refering to the response to Reviewer sj9U **Feedback 1**, we agree that our title suggested a broader reasoning scope than our experiments demonstrated. To better reflect our focus on spatial reasoning and disambiguation, we will revise the title to:
> > > “CoT-Spatialized Diffusion: Let's Reinforce T2I Generation Step-by-step”
> > > We believe this new title is more precise and better aligns with our core contribution.
> > >
> > > > **Weakness 3**: Errors in early stages (misinterpretation of spatial prompts or poor initial layout) could propagate and negatively affect the final output. The paper does not sufficiently analyze failure modes or the model’s ability to recover from such errors.
> > >
> > > **Answer of Review XeQc's W3:**
> > > While CoT-Diff operates as a multi-step cascade with theoretical error propagation risks, its core iterative feedback mechanism is inherently designed for self-correction.
> > >
> > > Even if an error occurs at step t, the MLLM re-evaluates and corrects it at step t+1. Figure 2 in the main text serves as a prime case study of this self-correction mechanism. In that example:
> > >
> > > 1. The MLLM initially generates a flawed 3D layout, misinterpreting the spatial relationship.
> > >
> > > 2. This leads to an incorrect intermediate generation (Image 1), where the relationship between woman and desk is wrong.
> > >
> > > 3. Crucially, in the next feedback loop, the MLLM assesses Image 1, detects the discrepancy between the generated image and the text prompt, and rectifies its own initial layout plan.
> > >
> > > 4. This corrected layout then guides the diffusion model to produce a spatially accurate intermediate image (Image 2), leading to a successful final output (Image N).
> > >
> > > Referring to the performance table for Reviewer XeQc **Q3**, the experimental results empirically validate that performance consistently improves with more evaluation and correction steps. This demonstrates our framework is a robust refinement mechanism, not a brittle cascade.
> > >
> > > While robust, failures can occur if the MLLM repeatedly makes incorrect assessments or if the diffusion model fails to follow the guidance. We will add a detailed qualitative analysis of these failure cases to the revised manuscript to provide a more complete picture.

---

### Official Review · Reviewer_UhfL · 2025-07-02

**Clarity:** 2
**Significance:** 2
**Originality:** 2
**Rating:** 4
**Confidence:** 4

**Summary:**

This paper presents CoT-Diff, a 3D-aware text-to-image generation pipeline. It uses a multimodal LLM to plan scene layouts and provide feedback at each diffusion step. Local prompt details and depth cues are injected via a condition-aware attention mechanism. The method is evaluated on three benchmarks including the newly introduced 3DSceneBench, and DVMP [30] and T2I-CompBench [14, 15].

**Questions:**

- How does performance change if you omit the custom attention mask?
- In Section 4.5 you mention comparing runtime efficiency, but the table omits these numbers. How does CoT-Diff’s inference time compare to other methods?

**Ethical Concerns:**

["NO or VERY MINOR ethics concerns only"]

**Final Justification:**

After reading the rebuttal and the reviews from other reviewers, and considering the discussion phase, most of my concerns have been addressed. The paper proposes a 3D-aware text-to-image generation pipeline that achieves state-of-the-art performance with relatively fast speed compared to other CoT works. Therefore, I recommend acceptance.

**Limitations:**

Yes

**Paper Formatting Concerns:**

In Table 1, “% Rareness” should be “% Complex.”

**Quality:**

2

**Strengths And Weaknesses:**

**Strengths**
- Well-written and easy to follow.
- Visual outputs are noticeably improved over prior methods.
- 3DSceneBench fills an important gap for evaluating 3D compositionality.

**Weaknesses**
- The authors do not compare against unified understanding-and-generation models such as Janus-Pro [*1] and ViLA-U [*2], which often performs well at text-image alignment. Including these baselines would provide useful insight.
- As the authors discussed, CoT-Diff will cause high computational overhead: using GPT-4o or Gemini 2.5 Pro at each step is time-consuming.
- The pipeline relies on intermediate images for feedback, but no visualizations of these images. The early steps are usually very blurry, leaving it unclear how useful they are for the MLLM.
- The proposed condition-aware attention mechanism closely resembles the one in EliGen, yet this similarity is not discussed.

[*1] Chen, Xiaokang, et al. “Janus-Pro: Unified Multimodal Understanding and Generation with Data and Model Scaling.” arXiv:2501.17811 (2025).

[*2] Wu, Yecheng, et al. “ViLA-U: A Unified Foundation Model Integrating Visual Understanding and Generation.” ICLR 2025.

---

> ### Author Rebuttal · Authors · 2025-07-30
>
> We sincerely thank you for the valuable feedback. Our main contribution is a CoT-lized framework that activates the powerful reasoning abilities of MLLMs to guide diffusion, effectively addressing the limitations of generative models in spatial reasoning and the rigidity of one-time layout planning.
>
>
> ## Weakness
>
> > **Weakness 1**: Lack of comparison with unified understanding-and-generation models.
>
> **Answer:**
> We have added comparisons with Janus-Pro[1], VILA-U[2], Image-CoT[3], and T2I-R1[4]. The results demonstrate CoT-Diff’s superior performance in spatial reasoning and text-image alignment, suggesting that current unified generation-understanding models still lack explicit reasoning capabilities compared to our CoT-lized framework.
>
> | Method      | Front | Behind | Front Left | Front Right | Back Left | Back Right | multi | complex |
> | :---------- | :---- | :----- | :--------- | :---------- | :-------- | :--------- | :---- | :------ |
> | janus-pro   | 35.2  | 36.5   | 29.7       | 24.9        | 24.9      | 22.9       | 19.6  | 39.0    |
> | vila-u      | 37.2  | 38.3   | 28.8       | 31.5        | 29.2      | 32.4       | 21.3  | 29.5    |
> | Image-CoT     | 48.2  | 49.7   | 56.1       | 61.2        | 56.7      | 57.6       | 39.7  | 46.6    |
> | t2i-r1      | 47.0  | 46.8   | 47.2       | 45.3        | 50.4      | 51.1       | 35.5  | 41.5    |
> | **CoT-Diff**    | **54.9**  | **55.2**   | **66.4**       | **69.0**        | **64.2**      | **64.5**       | **48.7**  | **50.8**    |
>
> > **Weakness 2**: As the authors discussed, CoT-Diff will cause high computational overhead: using GPT-4o or Gemini 2.5 Pro at each step is time-consuming.
>
> We thank the reviewer for this crucial point on computational overhead. We acknowledge that using a powerful MLLM introduces a time cost. To provide full transparency, we have conducted a detailed quantitative analysis.
> 1. Comparison with Existing Methods
> First, we compare CoT-Diff with both layout-based and other CoT-based methods. The baseline methods FLUX, RPG, and Eligen use 50 inference steps, and our CoT-Diff use 20 steps. The search number of Image-CoT is 20.
>
> | Method    | Planning Time | Feedback Time | Generation Time | Whole Time |
> |-----------|---------------|---------------|-----------------|------------|
> | FLux      | -             | -             | 28.1            | 28.1       |
> | RPG       | 19.8          | -             | 8.5             | 28.3       |
> | Eligen    | 5.2           | -             | 55.7            | 60.9       |
> | Image-CoT | -             | -             | 114.4           | 114.4      |
> | CoT-Diff  | 23.4          | 56.8          | 21.2            | 101.4      |
>
> The data shows that while CoT-Diff's Feedback Time adds to the overhead compared to simpler layout-based methods, its total inference time is comparable to other CoT-based frameworks like Image-CoT. This suggests the observed overhead is characteristic of the complexity required for step-by-step generation.
>
> 2. Adaptive Strategy for Efficiency
> More importantly, we have designed an adaptive early-stopping strategy to explicitly manage this cost without sacrificing quality. The MLLM guidance is not fixed but applied dynamically until the layout is deemed correct (up to a maximum of 5 steps).
> The table below shows how runtime scales with the number of MLLM feedback steps:
>
> | MLLM step | Planning Time | Feedback Time | Generation Time | Whole Time |
> |-----------|---------------|---------------|-----------------|------------|
> | base      | 23.4          | -             | 19.0            | 43.8       |
> | 1         | 23.4          | 19.3          | 19.5            | 62.2       |
> | 2         | 23.4          | 38.9          | 20.3            | 82.6       |
> | 3         | 23.4          | 58.8          | 21.4            | 102.5      |
> | 4         | 23.4          | 79.1          | 22.5            | 125.0      |
> | 5         | 23.4          | 99.2          | 23.3            | 145.9      |
> | ada (2.9) | 23.4          | 56.8          | 21.2            | 101.4      |
>
> Our experiments show that this adaptive strategy achieves considerable performance in just 2.9 steps on average, striking an effective balance between generation quality and computational cost.
>
> > **Weakness 3**: The pipeline relies on intermediate images for feedback, but no visualizations of these images. The early steps are usually very blurry, leaving it unclear how useful they are for the MLLM.
>
> **Answer:**
> We use FLUX-schnell as the diffusion backbone, which is a distilled version of FLUX-dev.
> Thanks to its improved denoising capability, even the early-stage intermediate images are relatively clean and informative.
> As shown in Figure 2 (Image 1 and Image 2), the visual quality at each step is sufficient for the MLLM to perform reliable evaluations and provide meaningful feedback.
>
> > **Weakness 4**: The proposed condition-aware attention mechanism closely resembles the one in EliGen, yet this similarity is not discussed.
>
>
> **Answer:**
> We thank the reviewer for pointing out the similarity between CoT-Diff's condition-aware attention mechanism and EliGen. We acknowledge that both methods indeed share common ground in their core mechanisms: both leverage global and local textual prompts to guide image content generation, and both employ similar isolation strategies by masking regions corresponding to different prompts (e.g., P1 and P2) during attention calculation to prevent semantic interference.
>
> However, CoT-Diff extends upon this foundation in two key aspects, endowing it with more powerful conditional control capabilities:
>
> 1) Introduction of Multimodal Conditions: Beyond textual prompts (such as p, p1, p2), CoT-Diff additionally incorporates a Depth Condition as a novel geometric input. This allows the model to simultaneously utilize both the semantic information from text and the precise spatial-geometric information from depth maps, enabling richer and more refined conditional control.
>
> 2) Extended Attention Mechanism for Heterogeneous Condition Fusion: To effectively fuse these two inherently heterogeneous conditional information types—text and depth—and to prevent potential conflicts and interference between them, we specifically extended the attention mechanism. By applying custom-designed attention masks, we ensure that information flow between different modal conditions (i.e., textual conditions and depth conditions) remains independent, thereby guaranteeing the precision and autonomy of each guiding information type.
>
>
> ## Question
>
> > **Question 1**:
> How does performance change if the custom attention mask is omitted?
>
> **Answer:**
> We have demonstrated the performance of the model without optimization. It can be observed that omitting the custom attention mask leads to a significant performance degradation in dimensions other than "front" and "behind". This is because the absence of the mask causes conditions from different objects to act on the same region, leading to issues such as attribute leakage and confusion.
>
> | Dataset        | Front | Behind | Front Left | Front Right | Back Left | Back Right | Multi | Complex |
> |----------------|-------|--------|------------|-------------|-----------|------------|-------|---------|
> | w/o attn       | 48.8  | 50.9   | 36.9       | 41.3        | 39.9      | 36.9       | 32.0  | 46.3    |
> | + attn         | 51.3  | 51.2   | 62.2       | 61.4        | 59.9      | 56.7       | 41.7  | 48.5    |
> | + optim (Full) | 54.9  | 55.3   | 66.4       | 69.0        | 64.1      | 64.5       | 48.7  | 50.8    |
>
>
>
> > **Question 2**:
> Section 4.5 mentions runtime efficiency comparison, but the table lacks these numbers; how does CoT-Diff’s inference time compare to other methods?
>
> **Answer:**
> As stated in W2, we adopt a quantitative comparison. The result shows CoT-Diff’s runtime is acceptable when compared against layout-based methods and is comparable to other CoT-based frameworks.
>
>
> [1] Chen, Xiaokang, et al. "Janus-pro: Unified multimodal understanding and generation with data and model scaling." arXiv preprint arXiv:2501.17811 (2025).
>
> [2] Wu, Yecheng, et al. "Vila-u: a unified foundation model integrating visual understanding and generation." arXiv preprint arXiv:2409.04429 (2024).
>
> [3] Guo, Ziyu, et al. "Can We Generate Images with CoT? Let's Verify and Reinforce Image Generation Step by Step." arXiv preprint arXiv:2501.13926 (2025).
>
> [4] Jiang, Dongzhi, et al. "T2i-r1: Reinforcing image generation with collaborative semantic-level and token-level cot." arXiv preprint arXiv:2505.00703 (2025).

---

> ### Author Response · Authors · 2025-08-03
> **Looking forward to further discussions**
>
> Dear Reviewer UhfL,
>
> We would like to sincerely thank you again for your insightful review comments.
>
> To address the main concerns you raised, we have provided a detailed response. Specifically, we have now included comprehensive comparisons with state-of-the-art unified models, conducted a thorough analysis of the computational overhead, and further clarified the novelty of our condition-aware attention mechanism. We hope these additions and clarifications have resolved your questions.
>
> As the author-reviewer discussion period is ending soon, we would be very grateful if you could take a moment to look over our rebuttal. Please let us know if our responses have adequately addressed your concerns.
>
> Thank you again for your time and valuable feedback.

---

> > ### Comment · Reviewer_UhfL · 2025-08-03
> > **Thanks for your response**
> >
> > Thanks for the authors detailed response. Most of my concerns are solved and I am willing to raise to score to 4.

---

> > > ### Author Response · Authors · 2025-08-04
> > >
> > > Thank you very much for your positive feedback and raising the score. We are glad our response addressed your concerns.

---

### Official Review · Reviewer_XeQc · 2025-07-06

**Clarity:** 3
**Significance:** 2
**Originality:** 2
**Rating:** 4
**Confidence:** 4

**Summary:**

Current text-to-image models often fail to accurately depict complex scenes with specific spatial layouts. The CoT-Diff framework addresses this by integrating a Multimodal Large Language Model (MLLM) with a diffusion model to enable step-by-step, "Chain-of-Thought" style reasoning during image creation. At each step of the generation process, the MLLM assesses the intermediate image, dynamically refines a 3D scene plan, and guides the diffusion model. This updated 3D layout is translated into semantic and depth map conditions, which are injected into the model via a condition-aware attention mechanism for precise control. To test this method, the researchers developed a new dataset, 3DSceneBench, which features prompts with complex 3D spatial relationships. Experiments demonstrate that CoT-Diff significantly improves spatial accuracy and compositional fidelity, outperforming state-of-the-art baselines by up to 34.7% in complex scenes.

**Questions:**

1. While Table 6 indicates a higher success rate for a "reasonable inference time," could you provide a more direct, quantitative comparison of the wall-clock time and computational cost (e.g., in GPU hours or API costs) per image for CoT-Diff versus the baselines like FLUX and EliGen? Furthermore, have you explored the trade-off between performance and efficiency by reducing the frequency of MLLM feedback (e.g., invoking the MLLM every N steps instead of every step)? Is continuous feedback essential, or can similar results be achieved with a more sparse feedback schedule?
2. How does CoT-Diff perform when paired with smaller, open-source MLLMs (e.g., Qwen family)? Does the performance degrade gracefully, or does the entire framework collapse without a state-of-the-art reasoning engine?
3. Could you provide a qualitative analysis of the system's failure modes? For instance, what happens if the MLLM makes an incorrect assessment of an intermediate image or misinterprets a spatial relationship early in the process? Does the system typically recover, or does the error propagate and lead to a completely failed generation?

**Ethical Concerns:**

["NO or VERY MINOR ethics concerns only"]

**Final Justification:**

The rebuttal responses almost addressed my concerns, so I remain my positive score.

**Limitations:**

Please refer to the aforementioned weaknesses.

**Quality:**

3

**Strengths And Weaknesses:**

Strengths:
1. The paper's primary strength is its core idea of "CoT-lized Diffusion," a well-engineered technical approach that tightly integrates MLLM-driven reasoning within the diffusion sampling loop. Unlike prior works that use an LLM for a one-time layout plan before generation begins, CoT-Diff creates a dynamic feedback system. The MLLM inspects intermediate images at each denoising step and refines the 3D layout on the fly. This represents a significant conceptual shift from static, decoupled pipelines to an entangled, step-by-step generation paradigm, which is a novel and valuable contribution to the field.
2. The paper's claims are supported by an exceptionally thorough evaluation. The experimental comparison against a strong suite of baselines, including both standard diffusion models and other layout-guided methods, shows state-of-the-art performance, with a particularly impressive 34.7% improvement in complex scene accuracy.

Weaknesses:
1. The framework's main drawback is its significant computational overhead. The core mechanism requires invoking a powerful MLLM at each denoising step to evaluate the image and refine the layout. While the paper claims "reasonable inference time", this iterative reasoning process is inherently much slower and more expensive than methods that perform layout planning only once. The authors acknowledge this limitation in its Appendix, but its severity makes the method impractical for many real-world applications and raises serious questions about scalability.
2. The system's "reasoning" capability is entirely outsourced to a large, proprietary MLLM. This creates several issues. First, it may harm the reproducibility because the performance is dependent on a black-box API that can change over time. Second, it limits the research contribution by not integrating the reasoning capacity directly into the generative model itself, a point the authors concede is future work. The success of CoT-Diff is therefore heavily contingent on the quality of an external, pre-existing model, rather than being a fully self-contained advancement.
3. The framework operates as a multi-step cascade, where an error at any stage can potentially corrupt the entire output. If the MLLM initially misinterprets the spatial relationships in the prompt or incorrectly evaluates an intermediate image, the subsequent layout refinements may be misguided. While the results are impressive, the paper does not sufficiently analyze the system's failure modes or its robustness to such errors. A single flawed "thought" in the chain could lead the generation process astray, and it is unclear if the system can reliably recover.

---

> ### Author Rebuttal · Authors · 2025-07-29
>
> We sincerely thank you for your positive and encouraging feedback. We are glad that you recognize the novelty of our CoT-lized Diffusion framework and its contribution to integrating MLLM-driven reasoning into the diffusion process. Based on your suggestions, we have incorporated several additional experiments into the paper to further showcase the effectiveness of our approach.
>
> ## Weakness
>
> > **Weakness 1**: The framework's main drawback is its significant computational overhead. The core mechanism requires invoking a powerful MLLM at each denoising step to evaluate the image and refine the layout.
>
> **Answer:**
> We thank the reviewer for their concerns regarding computational overhead. CoT-Diff's MLLM is not invoked at every denoising step; its primary cost is concentrated in the initial "thinking" phase for layout refinement.
>
> Inspired by this, we designed an adaptive early-stopping strategy: the MLLM ceases evaluation once the layout is deemed correct (max 5 steps), averaging just 2.9 steps. This effectively balances performance and efficiency, as shown in the second time cost table.
>
> | MLLM step | Planning Time | Feedback Time | Generation Time | Whole Time |
> |-----------|---------------|---------------|-----------------|------------|
> | base(0)   | 23.4          | -             | 19.0            | 43.8       |
> | 1         | 23.4          | 19.3          | 19.5            | 62.2       |
> | 2         | 23.4          | 38.9          | 20.3            | 82.6       |
> | 3         | 23.4          | 58.8          | 21.4            | 102.5      |
> | 4         | 23.4          | 79.1          | 22.5            | 125.0      |
> | 5         | 23.4          | 99.2          | 23.3            | 145.9      |
> | ada (2.9) | 23.4          | 56.8          | 21.2            | 101.4      |
>
> We argue this front-loaded computational cost is a reasonable trade-off for the significant improvements in generation quality and spatial accuracy. CoT-Diff's value in layout-critical applications like game asset creation, virtual staging, and film pre-visualization justifies this expense.
>
>
> > **Weakness 2**: The system's "reasoning" capability is entirely outsourced to a large, proprietary MLLM.
>
> **Answer:**
> We acknowledge CoT-Diff's reliance on external MLLMs, potentially impacting reproducibility. However, our core contribution lies in designing a novel framework that effectively bridges MLLM's high-level cognitive reasoning with diffusion models' pixel synthesis. Our key innovation demonstrates how to activate and dynamically inject powerful spatial and logical reasoning from external MLLMs into the diffusion process in a step-by-step, corrective manner, representing a significant advance in controllable generation. Furthermore, our framework is model-agnostic, and while we showcased maximum potential with a proprietary model, CoT-Diff can readily adapt to powerful open-source MLLMs (see Q2 response), solidifying its reproducibility.
>
>
> > **Weakness 3**: Lacking a sufficiently analysis for the system's failure modes or its robustness to such errors.
>
> **Answer:**
> While CoT-Diff operates as a multi-step cascade with theoretical error propagation risks, its core iterative feedback mechanism is inherently designed for self-correction.
>
> Even if an error occurs at step t, the MLLM re-evaluates and corrects it at step t+1. A prime example is Figure 2 in our main paper, where the MLLM's initial flawed layout (Image 1) is detected and rectified in the subsequent loop, leading to a corrected generation (Image 2) and a successful final output. The system is designed to recover from its own intermediate mistakes.
>
> We provide a detailed qualitative analysis and case study in our Q3 response explicitly demonstrating the system's recovery process, and experimental results empirically validate that increasing MLLM evaluation and correction steps consistently improves performance. This indicates CoT-Diff is not a brittle cascade but a robust mechanism leveraging its iterative nature for error correction and progressive refinement.
>
>
> ## Question
>
> > **Question 1**:
> Could you quantify CoT-Diff's inference time and computational cost against baselines (e.g., FLUX, EliGen)? Have you explored the trade-off between performance and efficiency by reducing MLLM feedback frequency, and is continuous feedback essential?
>
> **Answer:**
> We will include a detailed quantitative comparison of inference time and cost against baselines in the revised manuscript. The results are presented in the table below. Our results show that while CoT-Diff introduces overhead, our adaptive strategy keeps it manageable. The baseline methods FLUX, RPG, and Eligen use 50 inference steps, and our CoT-Diff use 20 steps. The search number of Image-CoT is 20.
>
> | Method    | Planning Time | Feedback Time | Generation Time | Whole Time |
> |-----------|---------------|---------------|-----------------|------------|
> | FLux      | -             | -             | 28.1            | 28.1       |
> | RPG       | 19.8          | -             | 8.5             | 28.3       |
> | Eligen    | 5.2           | -             | 55.7            | 60.9       |
> | Image-CoT | -             | -             | 114.4           | 114.4      |
> | CoT-Diff  | 23.4          | 56.8          | 21.2            | 101.4      |
>
> We believe that continuous feedback, especially early on, is crucial for timely and accurate layout correction. This is because the initial denoising steps establish the foundational structure of the image, where any modifications have a far more significant and decisive impact on the final output.
> A sparse feedback schedule (e.g., every N steps) risks missing critical windows for correction, potentially leading to errors that cannot be fixed in later stages. Therefore, our adaptive approach, which focuses intense guidance at the beginning and stops when the layout is stable, strikes a vital balance between ensuring high-quality results and managing computational cost.
>
> > **Question 2**:
> How does CoT-Diff perform when paired with smaller, open-source MLLMs (e.g., Qwen family)?
>
> **Answer:**
> We thank the reviewer for this important question regarding the framework's robustness with different MLLMs. We compared CoT-Diff's performance when guided by proprietary Gemini-2.5-Pro versus open-source Qwen2.5-VL-72B-Instruct, with results summarized below:
>
> | Model  | Front | Behind | Front Left | Front Right | Back Left | Back Right | Multi | Complex |
> |--------|-------|--------|------------|-------------|-----------|------------|-------|---------|
> | gemini | 54.9  | 55.3   | 66.4       | 69.0        | 64.1      | 64.5       | 48.7  | 50.8    |
> | qwen   | 48.3  | 49.1   | 59.2       | 61.4        | 58.3      | 58.2       | 42.2  | 49.0    |
>
> The results clearly indicate that the framework's performance degrades gracefully when switched to the open-source MLLM; critically, the system does not collapse and still outperforms other baseline methods. This demonstrates our framework's inherent value and robustness, supporting our W2 response that our approach is not fundamentally tied to a single proprietary MLLM and ensures reproducibility.
>
> > **Question 3**:
> Could you provide a qualitative analysis of the system's failure modes?
>
> **Answer:**
>
> We thank the reviewer for their question on system failure modes. CoT-Diff's core strength lies in its inherent self-correction capability.
>
> 1) **Qualitative Analysis:** Our framework is inherently designed to recover from initial errors. Figure 2 in the main text serves as a prime case study of this self-correction mechanism.
>
> 	In that example:
>
> 	(1) The MLLM initially generates a flawed 3D layout, misinterpreting the spatial relationship.
>
> 	(2) This leads to an incorrect intermediate generation (Image 1), where the relationship between woman and desk is wrong.
>
> 	(3) Crucially, in the next feedback loop, the MLLM assesses Image 1, detects the discrepancy between the generated image and the text prompt, and rectifies its own initial layout plan.
>
> 	(4) This corrected layout then guides the diffusion model to produce a spatially accurate intermediate image (Image 2), leading to a successful final output (Image N).
>
> While the system is robust, it can occasionally fail if the MLLM repeatedly makes incorrect assessments or if the diffusion model fails to follow the guidance. We will add a detailed qualitative analysis of such failure cases to the revised manuscript.
>
> 2) **Quantitative Validation:**
> To validate this, we evaluated performance at varying optimization steps. As the table below shows, performance consistently improves with more steps, saturating around 3 steps.
>
> | Step         | Front | Behind | Front Left | Front Right | Back Left | Back Right | Multi | Complex |
> |--------------|-------|--------|------------|-------------|-----------|------------|-------|---------|
> | base(0)      | 51.3  | 51.2   | 62.2       | 61.4        | 59.9      | 56.7       | 41.7  | 48.5    |
> | 1            | 53.3  | 51.9   | 63.4       | 66.0        | 61.2      | 59.8       | 45.8  | 48.4    |
> | 2            | 57.6  | 53.6   | 66.6       | 67.1        | 62.8      | 63.3       | 47.9  | 50.4    |
> | 3            | 55.2  | 54.5   | 67.5       | 69.5        | 62.2      | 64.9       | 47.6  | 49.7    |
> | 4            | 56.0  | 54.5   | 68.3       | 69.9        | 61.9      | 64.9       | 48.8  | 51.4    |
> | 5            | 56.2  | 56.4   | 67.9       | 71.2        | 64.4      | 66.1       | 51.0  | 51.8    |
> | ada (2.9)    | 54.9  | 55.3   | 66.4       | 69.0        | 64.1      | 64.5       | 48.7  | 50.8    |
>
> This trend indicates that the iterative process is effective at correcting errors and that the system reliably recovers and improves with more "thinking" time.

---

> > ### Comment · Reviewer_XeQc · 2025-08-08
> >
> > The rebuttal responses almost addressed my concerns, so I remain my positive score.

---

> > > ### Author Response · Authors · 2025-08-08
> > >
> > > Thank you very much for your positive feedback. We are glad our response addressed your concerns.

---

### Author Response · Authors · 2025-08-09
**Thanks for all ACs' and Reviewers' efforts**

Dear AC and all reviewers:

We sincerely appreciate your time and efforts in reviewing our paper. We are glad to find that reviewers recognized the following advantages of our work:

*   **Novelty (XeQc, QBNx, sj9U):** Reviewers highlighted the novelty of our 'CoT-lized' method, which introduces a dynamic, step-by-step reasoning paradigm by tightly integrating MLLM-driven 3D layout planning with the diffusion process.

*   **Strong Performance (XeQc, UhfL, sj9U):** The significantly improved visual outputs and state-of-the-art performance of our model, demonstrated by a thorough evaluation, were consistently noted as a key strength.

*   **New Benchmark (UhfL, QBNx):** The creation of our 3DSceneBench benchmark was recognized as a valuable and substantial contribution, filling an important gap for evaluating 3D compositional generation.

We also thank all reviewers for their insightful questions and constructive suggestions, helping further improve our paper. We summarize the major points of our response as follows:

*   **Comprehensive Experiments (UhfL, QBNx, sj9U):** In response to reviewer feedback, we have conducted additional experiments, comparing CoT-Diff with unified models (Janus-Pro, VILA-U), other image-cot generation methods (Image-CoT, T2I-R1), and a strong training-free baseline on the same FLUX architecture (RAG-Diffusion). We also included a human evaluation study. These results further validate the superior performance of our approach.

*   **Efficiency and Robustness (XeQc, UhfL, QBNx):** To address concerns about computational cost, a detailed inference time analysis was conducted across different methods and MLLM steps. This analysis validates that our adaptive early-stopping strategy strikes an effective balance between performance and efficiency, achieving a runtime comparable to competing approaches. Furthermore, we demonstrated our framework's robustness and reproducibility by achieving strong performance with a powerful open-source MLLM.

*   **In-depth Analysis (XeQc, UhfL, QBNx, sj9U):** We have provided deeper analysis on key aspects of our work. This includes the self-correction capabilities of system, detailed analysis of failure cases and trade-offs of fine-grained action generation versus global spatial composition. We have also committed to revising our paper’s title to "CoT-Spatialized Diffusion" to more precisely reflect its focus on spatial reasoning.

We are encouraged that our detailed responses have addressed the reviewers' concerns, leading to very positive feedback from all. These suggestions will be incorporated into the revised manuscript. Again, we sincerely thank all reviewers for their efforts and time.

Best,

Authors

---

### Note · Authors · 2025-08-12

We sincerely thank all reviewers and the AC for their constructive feedback and positive evaluations. In the rebuttal, we carefully addressed each concern by:

1. Extending experiments: adding comparisons with additional baselines, unified models, and image-CoT methods, as well as a human evaluation, all of which further confirmed the superiority of our approach.

2. Analyzing efficiency and robustness: providing detailed inference-time analysis, demonstrating the effectiveness of our adaptive early-stopping strategy, and showing reproducible strong results with open-source MLLMs.

3. Deepening analyses: offering self-correction capability studies, failure-case discussions, trade-off analyses, and clarifying our focus with a revised title (“CoT-Spatialized Diffusion”).

We are pleased that our detailed responses have successfully addressed the reviewers' concerns. The reviewers expressed satisfaction with the revisions, with some even raising their scores. We will incorporate all suggested improvements into the final version.

We truly appreciate the reviewers’ and AC’s time and effort, and are encouraged by their positive reception of our work.

The Authors

---

### Decision · Program_Chairs · 2025-09-17

**Decision:**

Accept (poster)

**Comment:**

Reviewers agree that this is a significant step forward in the field. The rebuttal and post-rebuttal discussion clarified initial questions. Congratulations, this submission is accepted.